# Enhancing bioreactor arrays for automated measurements and reactive control with ReacSight

François Bertaux [1,2,5,6✉], Sebastián Sosa-Carrillo[1,2,3,6], Viktoriia Gross [1,2,4], Achille Fraisse[1,2], Chetan Aditya [1,2,3], Mariela Furstenheim [1] & Gregory Batt [1,2✉]

Small-scale, low-cost bioreactors provide exquisite control of environmental parameters of microbial cultures over long durations. Their use is gaining popularity in quantitative systems and synthetic biology. However, existing setups are limited in their measurement capabilities. Here, we present ReacSight, a strategy to enhance bioreactor arrays for automated measurements and reactive experiment control. ReacSight leverages low-cost pipetting robots for sample collection, handling and loading, and provides a flexible instrument control architecture. We showcase ReacSight capabilities on three applications in yeast. First, we demonstrate real-time optogenetic control of gene expression. Second, we explore the impact of nutrient scarcity on fitness and cellular stress using competition assays. Third, we perform dynamic control of the composition of a two-strain consortium. We combine custom or chi.bio reactors with automated cytometry. To further illustrate ReacSight's genericity, we use it to enhance plate-readers with pipetting capabilities and perform repeated antibiotic treatments on a bacterial clinical isolate.

[1] Institut Pasteur, 28 rue du Docteur Roux, Paris, France. [2] Inria Paris, 2 rue Simone Iff, Paris, France. [3] Université Paris Cité, 85 boulevard Saint-Germain, Paris, France. [4] IAME Research Group, UMR 1137, Université Paris Cité and INSERM, Paris, France. [5] Present address: Lesaffre International, 101 rue de Menin, Marcq-en-Baroeul, France. [6] These authors contributed equally: François Bertaux, Sebastián Sosa-Carrillo. ✉email: f.bertaux@lesaffre.com; gregory.batt@inria.fr

Small-scale, low-cost bioreactors are emerging as powerful tools for microbial systems and synthetic biology research[1–4]. They allow tight control of cell culture parameters (e.g. temperature, cell density, media renewal rate) over long durations (several days). These unique features enable researchers to perform sophisticated experiments and to achieve high experimental reproducibility. Examples include characterization of antibiotic resistance when drug selection pressure increases as resistance evolves[1], cell-density controlled characterization of cell-cell communication synthetic circuits[2], and genome-wide characterization of yeast fitness under dynamically changing temperature using a pooled knockout library[3].

A weakness of existing small-scale, low-cost bioreactors is their limited automated measurement capabilities: in situ optical density measurements only inform about overall biomass concentration and its growth rate, and, when available[2,4], fluorescence measurements suffer from low sensitivity and high background. It is often essential to also measure and follow over time key characteristics of the cultured cell population, such as gene expression levels, cellular stress levels, cell size and morphology, cell cycle progression, proportions of different genotypes or phenotypes. Researchers usually need to manually extract, process and measure culture samples to run them through more sensitive and specialized instruments (e.g. a cytometer, a microscope, a sequencer). Manual interventions are usually tedious, error-prone and strongly constrains the available temporal resolution and scope (i.e. no time points during night-time). It also impedes the dynamic adaptation of culture conditions in response to such measurements. Such *reactive experiment control* is currently gaining interest in systems and synthetic biology. It can be used to either maintain a certain state of the population (external feedback control) or to maximize the value of the experiment (reactive experiment design). For example, external feedback control can be used to disentangle complex cellular couplings and signaling pathway regulations[5–8], to steer the composition of microbial consortia[9,10], or to optimize industrial bioproduction[11]. Reactive experiment design can be especially useful in the context of long and uncertain experiments such as artificial evolution experiments[12]. It is also useful to accelerate model-based characterization of biological systems by enabling real-time parameter inference and optimal experiment design[13].

In principle, commercial robotic equipment and/or custom hardware can be used to couple a bioreactor array to a sensitive, multi-sample (typically accepting 96-well plates as input) measurement device. However, this poses tremendous challenges regarding equipment sourcing, equipment cost, and software integration. When a functional platform is established, upgrade and maintenance of the corresponding hardware and software are also highly challenging. Accordingly, very few examples have been reported to date. For instance, only two groups have demonstrated automated cytometry and reactive optogenetic control of bacteria[14] or yeast[7] cultures, with setups limited to either a single continuous culture[14] or multiple cultures with limited continuous culture capabilities[7]. One group has also demonstrated automated microscopy and reactive optogenetic control of a single yeast continuous culture[15].

Here, we present ReacSight, a generic and flexible strategy to enhance bioreactor arrays for automated measurements and reactive experiment control. ReacSight is ideally suited to integrate open-source, open-hardware components but can also accommodate closed-source, GUI-only components (e.g. cytometers). First, we use ReacSight to assemble a platform enabling cytometry-based characterization and reactive optogenetic control of parallel yeast continuous cultures. Importantly, we build two versions of the platform, using either a custom-made bioreactor array or the recent low-cost, open-hardware, optogenetic-ready commercially available Chi.Bio bioreactors[4]. We then demonstrate its usefulness on three case studies. First, we achieve parallel real-time control of gene expression with light in different bioreactors. Second, we explore the impact of nutrient scarcity on fitness and cellular stress using highly controlled and informative competition assays. Third, we exploit nutrient scarcity and the reactive experiment control capabilities of the platform to achieve dynamic control over the composition of a two-strain consortium. Last, to further demonstrate the genericity of ReacSight, we use it to enhance a plate-reader with pipetting capabilities and to perform complex antibiotic treatments of an *E. coli* clinical isolate.

## Results

**Measurement automation, platform software integration, and reactive experiment control with *ReacSight*.** The ReacSight strategy to enhance bioreactor arrays for automated measurements and reactive experiment control combines hardware and software elements in a flexible and standardized manner (Fig. 1, Supplementary Note 1). A pipetting robot is used to establish, in a generic fashion, a physical link between any bioreactor array and any plate-based measurement device (Fig. 1a). Bioreactor culture samples are sent to the pipetting robot through pump-controlled sampling lines attached to the robot arm (*sampling*). A key advantage of using a pipetting robot is that diverse treatment steps can be automatically performed on culture samples before measurement (*treatment*). Samples are then transferred to the measurement device by the pipetting robot (*loading*). Naturally, this requires that the measurement device can be physically positioned such that when its loading tray is open, wells of the device input plate are accessible to the robot arm. Partial access to the device input plate is generally not a problem because the robot can be used to wash input plate wells between measurements, allowing re-use of the same wells over time (*washing*). Importantly, if reactive experiment control is not needed or if it is not based on measurements, the robot capabilities can also be used to treat and store culture samples for one-shot offline measurements at the end of an experiment, enabling automated measurements with flexible temporal resolution and scope.

ReacSight also provides a solution to several software challenges that should be addressed to unlock automated measurements and reactive experiment control of multi-bioreactors (Fig. 1b). First, programmatic control of all instruments of the platform (bioreactors, pipetting robot, measurement device) is required. Second, a single computer should communicate with all instruments to orchestrate the whole experiment. ReacSight combines the versatility and power of the Python programming language with the genericity and scalability of the Flask web application framework to address both challenges. Indeed, Python is ideally suited to easily build APIs to control various instruments: there exist well-established, open-source libraries for the control of micro-controllers (such as Arduinos), and even for the 'clicking'-based control of GUI-only software driving closed-source instruments lacking APIs (pyautogui). Importantly, the open-source, low-cost pipetting robot OT-2 (Opentrons) is shipped with a native Python API. Hamilton robots can also be controlled with a Python API[16]. Flask can then be used to expose all instrument APIs for simple access over the local network. The task of orchestrating the control of multiple instruments from a single computer is then essentially reduced to the simple task of sending HTTP requests, for example using the Python module requests. HTTP requests also enable user-friendly communication from the experiment to remote users using the community-level digital distribution platform Discord. This versatile instrument control architecture is a key component of ReacSight. Two other key components of ReacSight are (1) a generic object-oriented implementation of events (if *this* happens,

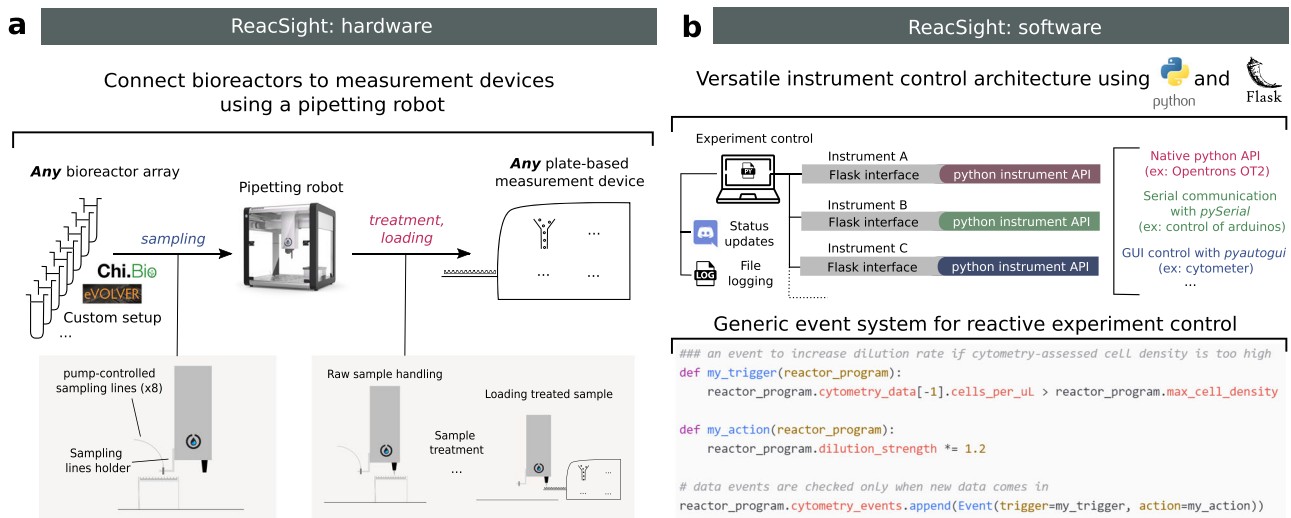

**Fig. 1 ReacSight: a strategy to enhance bioreactor arrays for automated measurements and reactive experiment control. a** On the hardware side, ReacSight leverages a pipetting robot (such as the low-cost, open-source Opentrons OT-2) to create a physical link between any multi-bioreactor setup (eVOLVER, Chi.Bio, custom…) and the input of any plate-based measurement device (plate reader, cytometer, high throughput microscope, pH-meter…). If necessary, the pipetting robot can be used to perform a treatment on bioreactor samples (dilution, fixation, extraction, purification…) before loading into the measurement device. If reactive experiment control is not needed, treated samples can also be stored on the robot deck for offline measurements (the OT-2 temperature module can help the conservation of temperature-sensitive samples). **b** On the software side, ReacSight enables full platform integration via a versatile instrument control architecture based on Python and the Python web application framework Flask. ReacSight software also provides a generic event system to enable reactive experiment control. Example code for a simple use case of reactive experiment control is shown. Experiment control can also inform remote users about the status of the experiment using Discord webhooks and generates an exhaustive log file.

do *this*) to facilitate reactive experiment control and (2) an exhaustive logging of all instrument operations into a single log file. ReacSight software as well as source files for hardware pieces are made openly available in the ReacSight Git repository.

**Reactive optogenetic control and single-cell resolved characterization of yeast continuous cultures.** Our first application of the ReacSight strategy is motivated by yeast synthetic biology applications. In this context, it is critical to accurately control synthetic circuits and to measure their output in well-defined environmental conditions and with sufficient temporal resolution and scope. Optogenetics provides an excellent way to control synthetic circuits, and bioreactor-enabled continuous cultures are ideal to exert tight control over environmental conditions for long durations. To measure circuit output in single cells, cytometry provides both high sensitivity and high throughput. We thus resorted to the ReacSight strategy to assemble a fully automated experimental platform enabling reactive optogenetic control and single-cell resolved characterization of yeast continuous cultures, using a benchtop cytometer as a measurement device (Fig. 2a).

Detailed information on the platform hardware and software is provided in Supplementary Note 2, and we discuss here only key elements. Eight reactors are connected to the pipetting robot, meaning that each time point fills one column of a sampling plate. While three columns of the cytometer input plate are accessible to the robot, we use only one column, washed extensively by the robot to achieve less than 0.2% carry-over, as validated using beads (Supplementary Fig. 8). We typically fit two tip boxes and two sampling plates ($2 \times 96 = 192$ samples) on the robot deck, therefore enabling 24 time points for each of the eight reactors without any human intervention. To enable reactive experiment control based on cytometry data, we developed and implemented algorithms to perform automated gating and spectral deconvolution between overlapping fluorophores (Fig. 2b, Supplementary Fig. 7).

We first validated the performance of the platform by carrying out long-term turbidostat cultures of yeast strains constitutively expressing various fluorescent proteins from chromosomally integrated transcriptional units (Fig. 2c). Distributions of fluorophore levels were unimodal and stable over time, as expected from steady growth conditions with a constitutive promoter. Distributions of mNeonGreen and mScarlet-I exactly overlapped between the single- and 3-color strains. This is consistent with the assumptions that expressing one or three fluorescent proteins from the strong pTDH3 promoter has negligible impact on cell physiology and that the relative positioning of transcriptional units in the 3-color strain (mCerulean first, followed by mNeonGreen and mScarlet-I) has little impact on gene expression. Measured levels of mCerulean appear slightly higher (~15%) in the 3-color strain compared to the single-color strain. This could be caused by residual errors in the deconvolution, exacerbated by the low brightness of mCerulean compared to autofluorescence and to mNeonGreen.

To validate the optogenetic capabilities of the platform, we built and characterized a light-inducible gene expression circuit based on the EL222 system[17] (Fig. 2d). As expected, applying different ON–OFF temporal patterns of blue light resulted in dynamic profiles of fluorophore levels covering a wide range, from near-zero levels (i.e., hardly distinguishable from autofluorescence) to levels exceeding those obtained with the strong constitutive promoter pTDH3 (Supplementary Fig. 9). Cell-to-cell variability in expression levels at high induction is also low, with coefficient of variation (CV) values comparable to the pTDH3 promoter (0.22 vs 0.20).

The first platform we assembled used a pre-existing, custom optogenetic-enabled bioreactor array (Supplementary Fig. 5). This setup has several advantages (reliability, wide range of working volumes) but cannot be replicated easily by other labs. Thanks to the modularity of the ReacSight architecture, we could quickly construct a second version of the platform with similar capabilities by exchanging this custom bioreactor array with an

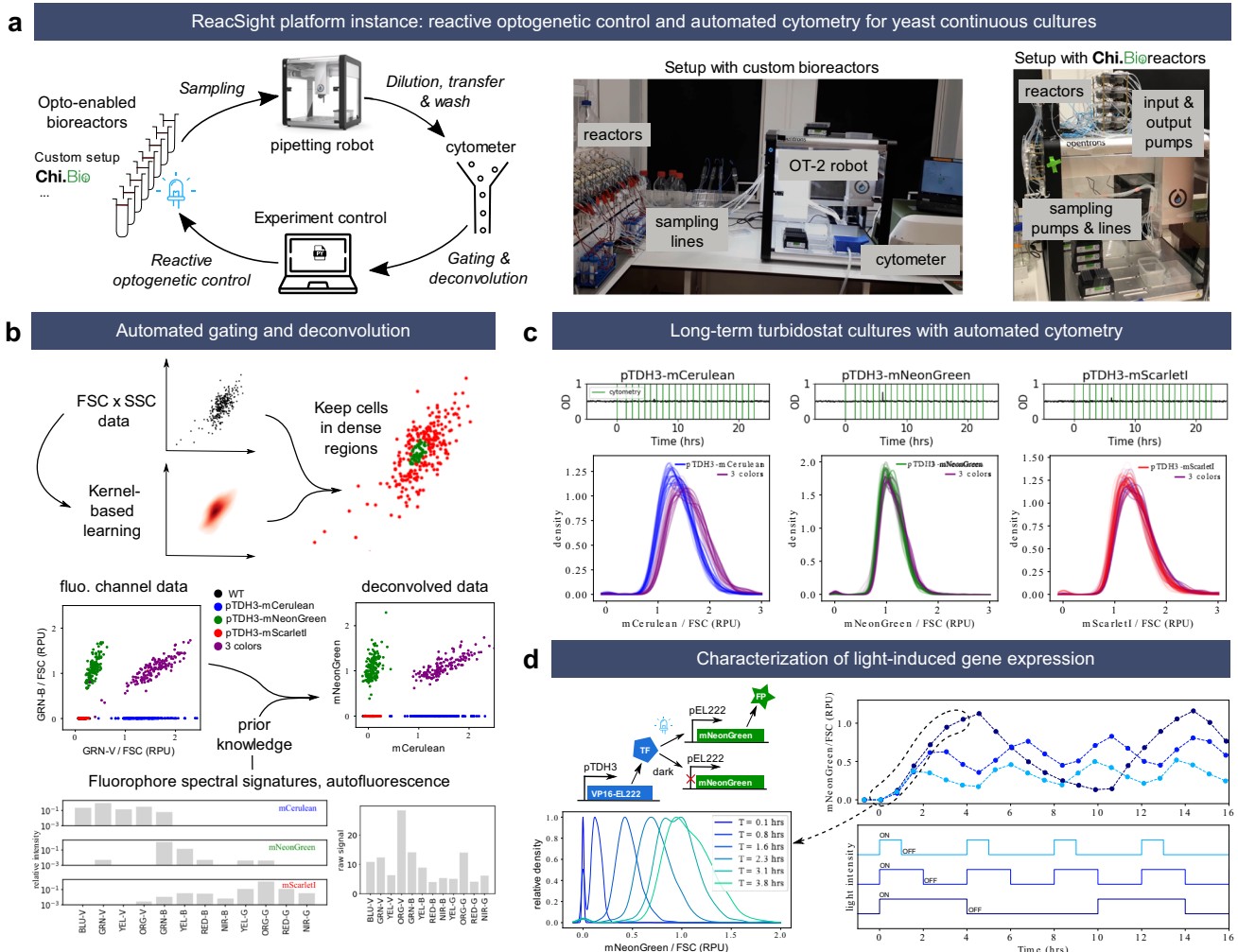

**Fig. 2 ReacSight-based assembly of an automated platform enabling reactive optogenetic control and single-cell resolved characterization of yeast continuous cultures. a** Platform overview. The Opentrons OT-2 pipetting robot is used to connect optogenetic-ready multi-bioreactors to a benchtop cytometer (Guava EasyCyte 14HT, Luminex). The robot is used to dilute fresh culture samples in the cytometer input plate and to wash it between time points. The 'clicking' Python library pyautogui is used to create the cytometer instrument control API. Custom algorithms were developed and implemented in Python to automatically gate and deconvolve cytometry data on the fly. Two versions of the platform were assembled, using either a custom bioreactor setup (left photo) or Chi.Bio reactors[4] (right photo). **b** Description of the gating and deconvolution algorithm. As an example, deconvolution between the overlapping fluorophores mCerulean and mNeonGreen are shown. **c** Stability of single-cell gene expression distributions over many generations. Strains constitutively expressing either mCerulean, mNeonGreen or mScarlet-I alone or altogether ('3-colors' strain) from the transcriptional units driven by the pTDH3 promoter and integrated in the chromosome were grown in turbidostat mode (OD setpoint = 0.5, upper plots) and cytometry was acquired hourly (vertical green lines). Distributions (smoothed via Gaussian kernel density estimation) of fluorophore levels (after gating, deconvolution, and normalization by the forward scatter, FSC) for all time points are plotted together with different color shades (bottom). RPU: relative promoter units (see Methods). The OD data for the '3-colors' are not shown for simplicity and are similar to the others. **d** Characterization of a light-driven gene expression circuit based on the EL222 system[17]. Three different ON–OFF blue light temporal profiles were applied (bottom) and cytometry was acquired every 45 min. The median of gated, deconvolved, FSC-normalized data is shown (top). All bioreactor experiments presented in this figure were performed in parallel, the same day, with the custom bioreactor platform version. Source data are provided as a Source Data file.

array of the recently described, open-hardware, optogenetic-ready, commercially available Chi.Bio[4] bioreactors (Fig. 2a (right photo), Supplementary Fig. 6). To validate the performance of this other version of the platform, we performed optogenetic induction experiments with the same strain as in Fig. 2d and obtained excellent reactor-to-reactor reproducibility for various light induction profiles (Supplementary Fig. 6).

**Real-time control of gene expression using light**. To showcase the reactive optogenetic control capabilities of the platform, we set out to dynamically adapt light stimulation so as to maintain fluorophore levels at different target setpoints. Such in-silico

feedback for in-vivo regulation of gene expression is useful to dissect the functioning of endogenous circuits in the presence of complex cellular regulations and could facilitate the use of synthetic systems for biotechnological applications[6,11,18].

We first constructed and validated a simple mathematical model of light-induced gene expression (Fig. 3a). Joint fitting of the three model parameters to the characterization data of Fig. 2d resulted in an excellent quantitative agreement. This is remarkable given the simplicity of the model assumptions: constant rate of mRNA production under light activation, constant translation rate per mRNA, and first-order decay for mRNA (mainly degradation, half-life of 20 min) and protein (mostly dilution, half-life of 1.46 h). Therefore, when experimental conditions are

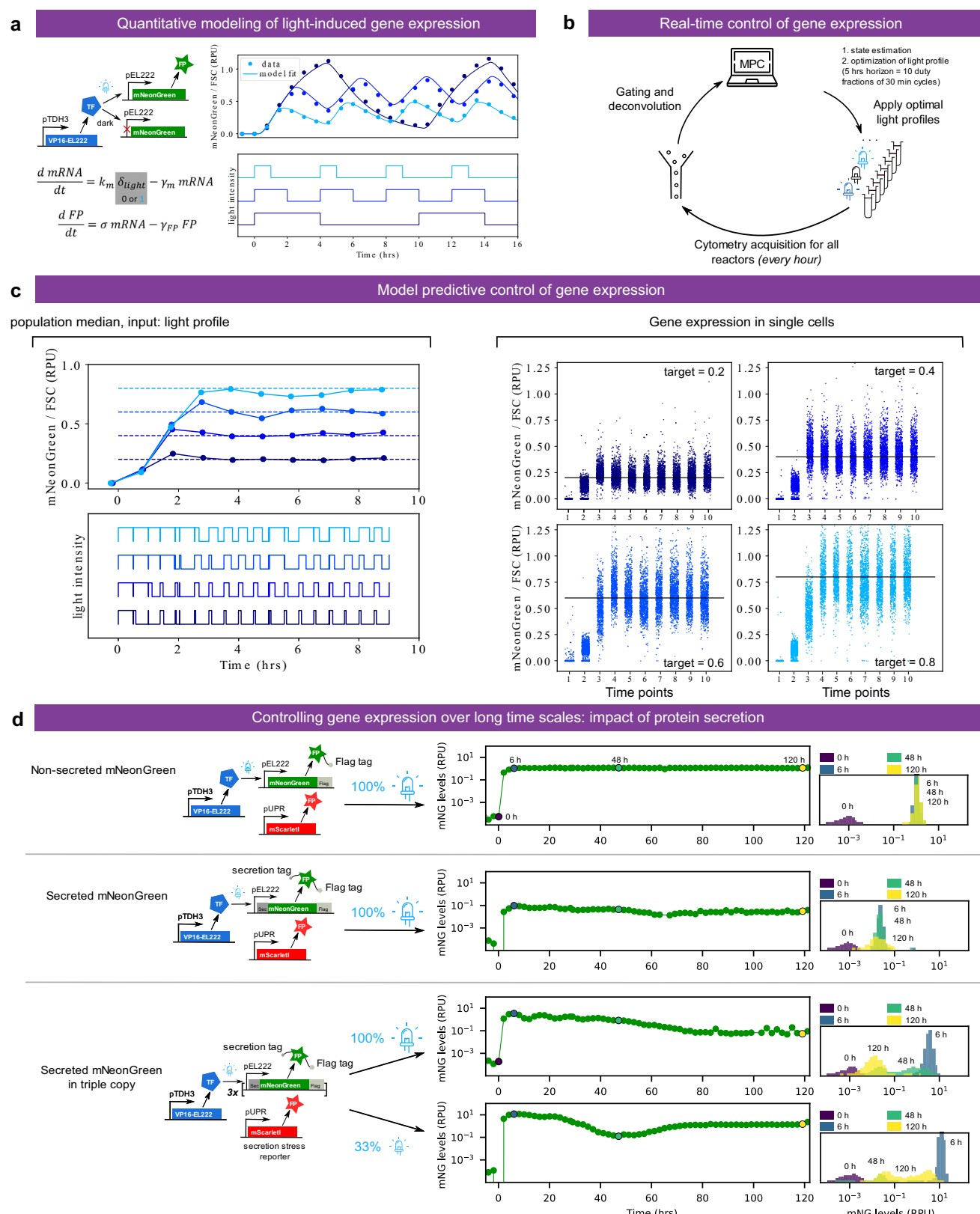

well-controlled and data are properly processed, one can hope to quantitatively explain the behavior of biological systems with a small set of simple processes. We then incorporated the fitted model into a model-predictive control algorithm (Fig. 3b). Together with the ReacSight event system, this algorithm enabled accurate real-time control of fluorophore levels to different targets

in different reactors in parallel (Fig. 3c). To further demonstrate the robustness and reproducibility of the platform, we performed several months later another single 8-reactor experiment involving quadruplicate reactor runs for two fluorophore target levels (Supplementary Fig. 10). All replicates achieved excellent tracking of the target, and the light profiles decided by the control

**Fig. 3 Closing the loop: real-time control of gene expression using light. a** A simple ODE model of the light-driven gene expression circuit is fitted to the characterization data of Fig. 2d. Fitted parameters are $\gamma_m = 2.09\,h^{-1}$, $\sigma = 0.64\,RPU\,h^{-1}$, and $\gamma_{FP} = 0.475\,h^{-1}$·$k_m$ was arbitrarily set to equal $\gamma_m$ to allow parameter identifiability from protein median levels only. **b** Strategy for the real-time control of gene expression. Every hour, cytometry acquisition is performed, and after gating, deconvolution, and FSC-normalization, data are fed to a model-predictive control (MPC) algorithm. The algorithm uses the model to search for the best sequence of duty fractions for 10 duty cycles of period 30 min (i.e. a receding horizon of 5 h) in order to track the target level. **c** Real-time control results for four different target levels, performed in parallel in different bioreactors (custom setup). Left: median of single cells (controlled value). Right: single-cell distributions over time. Note that a linear scale is used on all plots. **d** Long-term stability of the expression system and impact of protein secretion. Cells expressing an EL222-driven mNeonGreen fluorescent reporter, secreted or not, are grown in turbidostats for 5 days with cytometry measurements every 2 h. The mean expression level is represented for the entire duration of the experiment. Fluorescence distributions are also shown at selected time points (0, 6, 48, and 120 h after induction). Cells also harbor a fluorescent reporter for secretion stress (pUPR-mScarlet-I). Results are also provided for a secreted form of the mNeonGreen reporter protein integrated in three copies. The temporal evolution of the distributions of the protein of interest (mNeonGreen levels) and of the stress levels (mScarlet-I levels) are provided in Supplementary Figs. 11 and 12. Source data are provided as a Source Data file.

algorithms were highly similar, yet not identical, between replicates of the same target.

We also investigated the genetic stability of the induction system we used previously over longer time scales. Genetic stability is an important factor for industrial bioproduction[19,20]. We observed that the induction of the EL222-driven mNeon-Green protein can be sustained over 5 days with great stability (Fig. 3d top). Going further, we tested whether a secreted version of the same protein shows a comparable stability of expression. We observed that cellular levels were significantly lower and decreased after ~2 days of induction. Cellular heterogeneity increased as well (Fig. 3d right and Supplementary Fig. 11). In an attempt to compensate for the decrease in cellular levels, we integrated the expression cassette in multiple copies (three times, tandem chromosomic insertion). We obtained very high fluorescence levels after induction (Fig. 3d bottom). Surprisingly, these levels were an order of magnitude higher than for the non-secreted protein and were accompanied by an intense stress, as reported by an unfolded protein stress reporter (pUPR-mScarlet-I, Supplementary Fig. 12). After induction, intracellular protein levels gradually dropped. Intracellular protein levels showed clear bimodal distributions, strong indicators of genetic instability (Fig. 3d right and Supplementary Fig. 11). Lastly, the same triple-copy construct showed a non-monotonic behavior when induced at a third of the maximal induction level: a high initial response followed by a slow decrease in intracellular levels like the fully induced triple construction, followed by a non-expected slow recovery of high internal protein levels on the long-term (Fig. 3d bottom). This recovery could be explained either by cellular adaptation to high production demands or, more likely, by selection of the high producing subpopulation that better preserved the HIS3 selection marker conferring a slight growth advantage even in complete media. This experiment demonstrates the capability of our platform to perform long experiments and provide single-cell information with a relatively high temporal resolution. Moreover, it motivated us to explore and exploit the impact of nutrient availability on fitness and stress.

**Exploring the impact of nutrient scarcity on fitness and cellular stress.** Fluorescent proteins can be used as reporters to assess phenotypic traits of cells or as barcodes to label strains with specific genotypes[21]. Together with automated cytometry from bioreactor arrays, this capability extends the range of possible experiments: multiplexed strain characterization and competition in dynamically controlled environments (Fig. 4a). Indeed, some fluorescent proteins can be used for genotyping and others for phenotyping. Automated cytometry (including raw data analysis) will then provide quantitative information on both the competition dynamics between the different strains and cell-state

distribution dynamics for each strain. Depending on the goal of the experiment, this rich information can be fed back to experiment control to adapt environmental parameters for each reactor.

As a first proof of concept that such experiments can be carried out, we set out to explore the impact of nutrient scarcity on fitness and cellular stress (Fig. 4b, top-left). Different species in microbial communities have different nutritional needs depending on their metabolic diversity or specialization, and their fitness therefore depends not only on external environmental factors but also on the community itself through nutrient consumption, metabolite release, and other inter-cellular couplings[22,23]. As opposed to competition assays in batch, continuous culture allows to control for such factors. For example, in turbidostat cultures, nutrient availability depends on both nutrient supply (i.e. nutrient levels in the input medium) and nutrient consumption by cells (which primarily depends on the OD setpoint). We used histidine auxotrophy as a model for nutrient scarcity: for his3 mutant cells, histidine is an essential nutrient. By competing his3 mutant cells with wild-type cells at different OD setpoints and different histidine concentrations in the feeding medium, we can measure how nutrient scarcity affects fitness (Fig. 4b, top-right). Using a stress reporter in both strains also informs about the relationship between fitness and cellular stress in the context of nutrient scarcity. We focused on the unfolded protein response (UPR) stress[24] to investigate whether nutrient stress can lead to other, a priori unrelated types of stress, which will be indicative of global couplings in cell physiology.

At a histidine concentration of 4 μM, his3 mutant cells are strongly outcompeted by wild-type cells over the range of OD setpoints (0.1–0.8) we considered (Fig. 4b, bottom left). This is not the case anymore at a concentration of 20 μM. At this concentration, the growth rate advantage of wild-type cells is close to zero below an OD setpoint of 0.6 (the remaining histidine is sufficient for his3 mutant cells to grow normally) and becomes larger than $0.2\,h^{-1}$ at the largest OD setpoint of 0.8 (the remaining histidine is too low and limits growth of his3 mutant cells). Therefore, for this level of nutrient supply, levels of nutrient consumption by cells have a strong impact on fitness of his3 mutant cells. This qualitative change between 4 μM and 20 μM is highly consistent with the reported value of 17 μM for the $K_m$ constant of the single high-affinity transporter of histidine, HIP1[25]. Also, because the growth rate difference between wild-type and mutant cells for a histidine concentration of 4 μM is close or even exceeds the typically observed growth rate of wild-type cells (between 0.3 and $0.45\,h^{-1}$ depending on the OD setpoint), we conclude that mutant cells are fully growth-arrested in these conditions. UPR data show little difference between mutant and wild-type cells across all OD setpoints for a histidine concentration of 20 μM but a clear activation of the UPR response in mutant cells at a histidine concentration of 4 μM

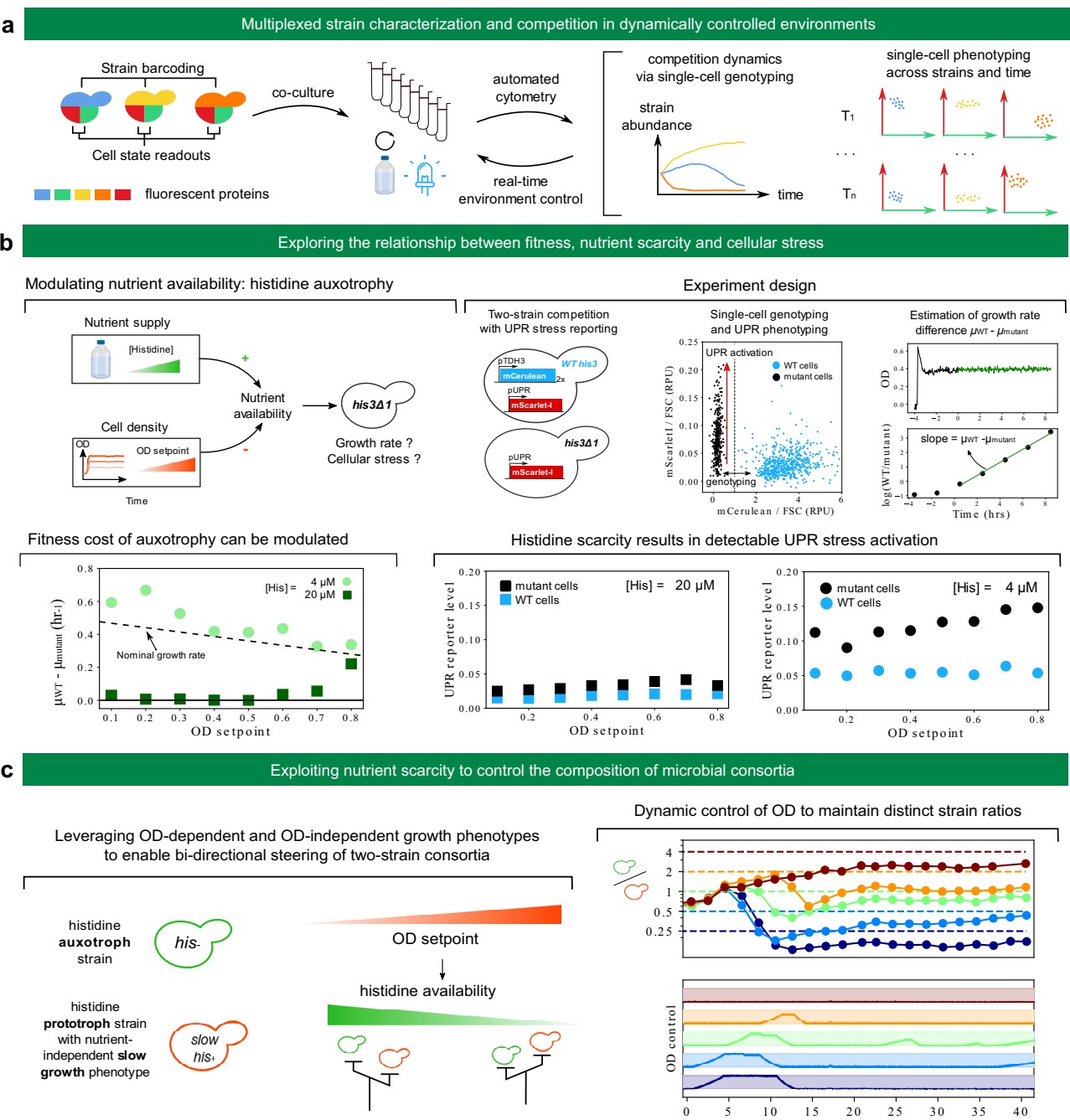

**Fig. 4 Exploring and exploiting the relationship between fitness, nutrient scarcity and cellular stress. a** Experiments combining single-cell genotyping and phenotyping are made possible thanks to co-cultures, automated cytometry and reactive experiment control to adapt environmental conditions in real-time. **b** Top-left: the availability of essential nutrients (such as histidine for his3 mutant strains) depends on the environmental supply but also on cell density via nutrient consumption. Low nutrient availability will impede growth rate and might trigger cellular stress. Top-right: experiment design. Wild-type cells (marked with mCerulean constitutive expression) are co-cultured with his3 mutant cells. Both strains harbor a UPR stress reporter construct driving expression of mScarlet-I. Automated cytometry enables to assign single cells to their genotype (Supplementary Fig. 13) and to monitor strain-specific UPR activation. The dynamics of the relative amount of the two strains allows inference of the growth rate difference between mutant and wild-type cells for each condition. Bottom left: cell density dependence of the fitness deficit of mutant cells at two different media histidine concentration. The dashed line indicates the approximate dependence of wild-type growth rate on the OD setpoint. Bottom-right: strain-specific UPR activation for each condition. **c** Left: principle for a two-strain consortium whose composition can be steered using control of OD. Right: implementation and demonstration. The secretion of a heterologous difficult-to-fold protein is used as a nutrient-independent slow-growth phenotype. Dynamic control of the OD setpoint is performed using model-predictive control and the ReacSight event system, similarly to Fig. 3b (see Methods). Blue light was started at time 0 and was kept ON during the whole experiment to induce the slow-growth phenotype of the slow his+ strain. We note the systematic presence of a steady-state error, with measured ratio below the target. In Supplementary Note 3, we investigate the mechanisms limiting the control performance (instability of the slow-growth phenotype, strain identification errors, and delays not accounted for in the model) and we also provide results of additional control experiments (Supplementary Figs. 14–17). Source data are provided as a Source Data file.

(Fig. 4b, bottom-right). Therefore, seemingly similar growth phenotypes (such as mutant cells at OD 0.8 for 4 and 20 μM) can correspond to different physiological states (as revealed by differences in UPR stress levels).

In addition, to showcase reactive control of the environment informed by strain abundance data, we set out to dynamically control the ratio of two strains. Taking control over the composition and heterogeneity of microbial cultures is anticipated to enable more efficient bioprocessing strategies[9,10,26]. We reasoned that the OD of the culture could be used as a steering knob when one of the two strain is auxotroph for histidine. Indeed, the strong OD-dependence of the histidine biosynthesis mutant growth rate at a medium histidine concentration of 20 μM (Fig. 4b, bottom left) means that switching the OD setpoint of turbidostat cultures can be used to dynamically control its growth rate. In addition, if such strain is co-cultured with a strain prototroph for histidine but growing slower in an OD-independent manner, bi-directional steering of the two strains ratio can be achieved (Fig. 4c, left). We built such strain by leveraging burdensome heterologous protein secretion. We then constructed a simple model to predict the (steady-state) growth rate difference with the histidine auxotroph strain. Using this model for model-predictive control and the ReacSight event system, we could maintain distinct ratios of the two strains in parallel bioreactors (Fig. 4c, right) in a fully automated fashion. We noted however the systematic presence of a steady-state error. This behavior was likely due to an unexpected recovery of the growth rate of the slow strain. Because this behavior has not been observed in characterization experiments, we hypothesized that this difference was due to the different composition of the amino-acid supply mixtures that were used in the characterization or control experiments (besides histidine, the histidine drop-out supplement of Sigma is richer that the complete supplement of Formedium). Additional characterization experiments and control results are provided in Supplementary Note 3 and in Supplementary Figs. 14–17.

**ReacSight is a generic strategy: enhancing plate readers with pipetting capabilities**. To illustrate the genericity of ReacSight as a strategy to create experimental platforms by connecting lab equipment to grow cells and/or measure cellular readouts together with pipetting robots, we have connected a Tecan plate reader with an Opentrons pipetting robot (Fig. 5a). The pipetting robot and the computer driving the plate reader are interfaced via Flask. Because we do not have access to an API for the plate reader, we used again a 'clicking'-based control strategy using pyautogui.

In a first application, we use the pipetting robot to maintain bacterial cell populations in growing conditions for extended periods of time. More specifically, an *E. coli* clinical isolate is grown in two different media (M9 glucose with or without casamino acids) and in presence of various concentrations of cefotaxime (CTX), a β-lactam antibiotic. The chosen isolate is resistant to cefotaxime treatments thanks to the expression of β-lactamases. It has a minimum inhibitory concentration to CTX of 2 mg/L. When the median of the cell population ODs reaches a target level, media is renewed following a strategy that compensates for evaporation (Fig. 5b left). With the chosen strategy, we were able to maintain the median OD close to the chosen target (0.05 or 0.1) for at least 15 cell generations (Fig. 5b right). Interestingly, we observed that cells resist better in glucose + casamino acids than in glucose alone when treated with 1 mg/L of cefotaxime. This is somewhat surprising since β-lactam antibiotics generally have a stronger impact on cells in fast-growing conditions[27,28].

In a second application, we used this platform to test the effect of a second dose of cefotaxime, applied at different cell densities. These experiments are conceptually very simple but their outcomes are highly challenging to predict. Low concentrations of cefotaxime inhibit the PBP3 proteins, involved in cell division, and thus lead to filament formation, whereas higher concentrations cause inhibition of the PBP1 proteins, involved in cell wall maintenance, and result in bacterial lysis[29–31]. Thanks to filamentation, exponential growth of the population biomass may continue during extended durations, even in absence of cell divisions. Moreover, β-lactamases released by dead cells degrade the antibiotic in the environment. This results in a race against the clock between cell death and antibiotic degradation, with filamentation contributing to delay this race and increase biomass in the meantime (Fig. 5c left). Therefore, experiments in which a second dose of antibiotics is applied at different cell densities have the potential to be enlightening to understand the different effects at play (Fig. 5c middle). When starting at an optical density of $5\ 10^{-4}$, results of single treatments were consistent with the MIC of the isolate since treatments above the MIC lead to a pronounced arrest of growth whereas treatment below the MIC did not (Fig. 5c, "treatment" with media). One can also observe that in the former case growth resumed after several hours, a behavior typical of enzyme-mediated antibiotic tolerance[32]. These two observations remained valid in the case of a second treatment with 16 mg/L of CTX. Interestingly, when growth stopped upon treatment, the OD at crash appeared to be approximatively 25 times higher than the OD at treatment: $12\ 10^{-3}$, $6\ 10^{-2}$, and $12\ 10^{-2}$, for treatments at $5\ 10^{-4}$, $2.5\ 10^{-3}$, and $5\ 10^{-3}$, respectively. This suggests that antibiotic degradation by live cells before the crash was limited, and consequently, that only a limited number of cells died before the crash. Therefore, tolerance to antibiotic treatments allowed cells to increase biomass almost 25 times before death, and then thanks to enzyme-mediated antibiotic degradation, survive treatments well above their MIC. One can also observe that the lag between the crash and regrowth was relatively constant (~5 h) when the initial treatment was 4 mg/L, irrespectively of the total amount of antibiotic added (4 or 20 mg/L CTX). This suggests that antibiotic degradation was very efficient after the crash and that the lag mainly corresponds to the time needed for non-detectable regrowth when the dynamics of live cells is hidden by the optical density of the dead biomass. In our conditions, when the first treatment is effective (4 or 16 mg/L), the second treatment appeared to have little to no effect. An in-depth study would be needed to investigate these effects in a more quantitative manner.

## Discussion

We report the development of ReacSight, a strategy to enhance multi-bioreactor setups with automated measurements and reactive experiment control. ReacSight addresses an unmet need by allowing researchers to combine the recent advances in low-cost, open-hardware instruments for continuous cultures of microbes (e.g. eVOLVER, Chi.Bio[3,4]) and multi-purpose, modular pipetting robots (e.g. Opentrons OT-2) with sensitive, but generally expensive, stand-alone instruments to build fully automated platforms that significantly broadens the set of feasible experiments. We also demonstrate that ReacSight can be used to enhance plate readers with pipetting capacities. ReacSight is generic and easy to deploy, and should be broadly useful for the microbial systems biology and synthetic biology communities.

As already noted by Wong and colleagues[3], connecting a multi-bioreactor setup to a cytometer for automated measurements could enable single-cell resolved characterization of microbial

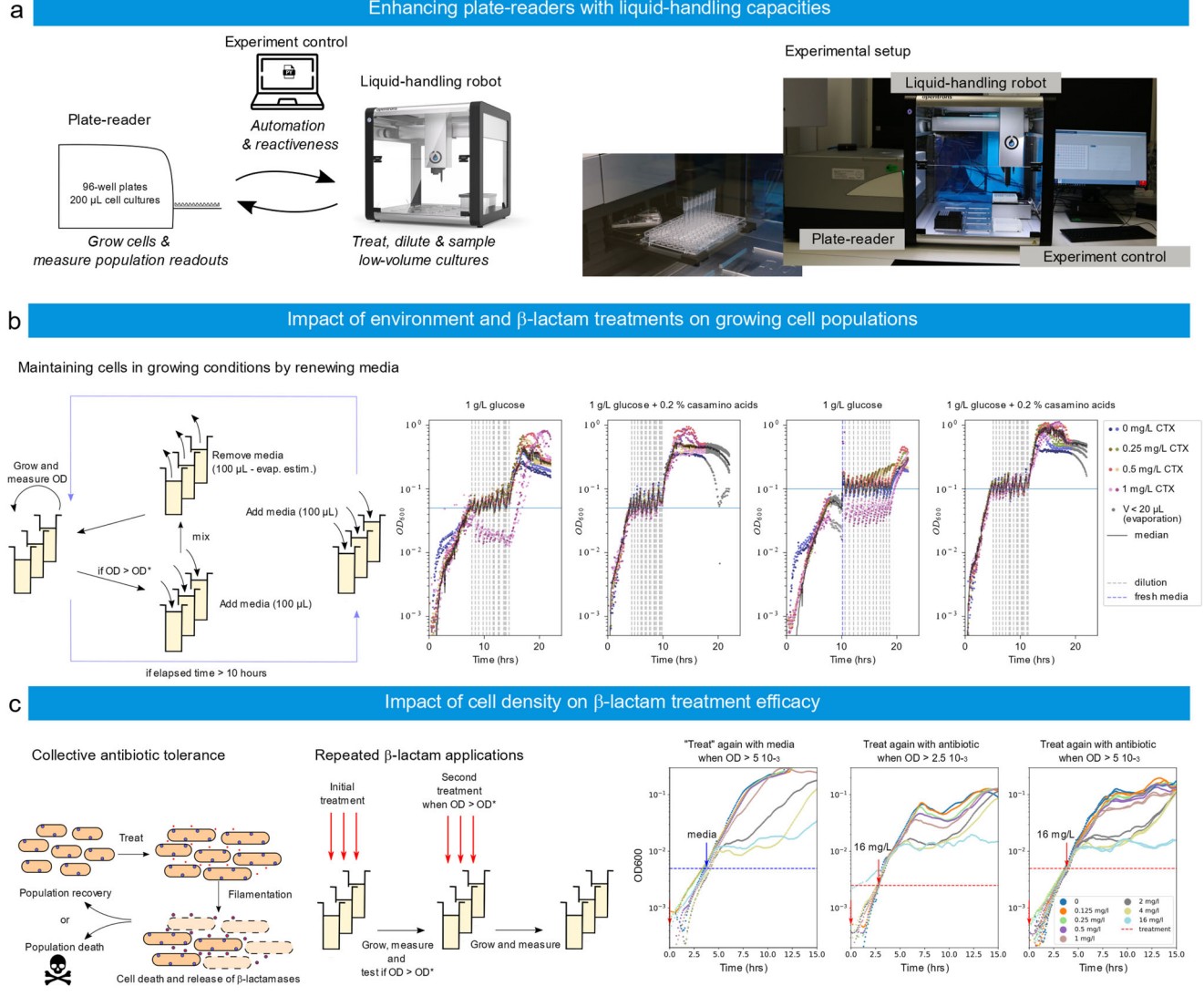

**Fig. 5 ReacSight-based assembly of an automated platform enabling reactive control and characterization of bacterial cultures in low-volumes.**
**a** Platform overview. The Opentrons OT-2 pipetting robot is used to enhance the capacities of a plate-reader (Spark, Tecan). The robot is used to treat cultures in the plate-reader at predefined ODs. **b** Left: an E. coli clinical isolate can be maintained in growing conditions by renewing the media in an OD-controlled manner. Care must be taken to compensate for evaporation over extended time scales. Right: cells in rich media (glucose + casamino acids vs glucose alone) grow faster and yet resist better sub-MIC antibiotic treatments. **c** Left: A bacterial population may exhibit resilience to treatments thanks to the combination of two effects. At the single-cell level, cells may tolerate an antibiotic concentration exceeding their MIC through filamentation. Filamentation-based tolerance allows to increase biomass before cell death. At the population level, the antibiotic is degraded by enzymes released upon cell death in the environment. The final outcome depends on a race between cell death and antibiotic degradation. Middle: the respective role of these two effects can be investigated by means of repeated antibiotic treatments. Right: an E. coli clinical isolate is treated with different concentrations of CTX (legend) at an initial of OD of $5\ 10^{-4}$, and a second time with either 16 mg/L of CTX (red) or media alone (blue) at a user-defined OD ($2.5\ 10^{-3}$ or $5\ 10^{-3}$). Because of instrument limitations, OD readouts below $10^{-3}$ are poorly reliable. Source data are provided as a Source Data file.

cultures across time. Automated cytometry in the context of microbial systems and synthetic biology has in fact already been demonstrated years ago by a small number of labs[6,14,33], but low throughput or reliance on expensive automation equipment likely prevented a wider adoption of this technology. Automated cytometry from continuous cultures becomes especially powerful in combination with recently developed optogenetic systems[34,35], enabling targeted, rapid and cost-effective control over cellular processes[14]. We used ReacSight to connect two distinct bioreactor setups (our own, pre-existing custom setup and the recent Chi.Bio[4] optogenetic-ready bioreactors) with a cytometer. This demonstrate the modularity of the ReacSight strategy, and the platform version using Chi.Bio bioreactors illustrates how other labs lacking pre-existing bioreactor setups could build such

platform at a small time and financial cost (excluding the cost of the cytometer, which are expensive but already widespread in labs given their broad usefulness even in absence of automation). We demonstrated the key capabilities of such platform by performing, in a fully automated fashion and in different reactors in parallel, (1) light-driven real-time control of gene expression; (2) cell-state informing competition assays in tightly controlled environmental conditions; and (3) dynamic control of the ratio between two strains.

Still, we only touched the surface of the large space of potential applications offered by such platforms. Strain barcoding can be scaled up to 20 strains with two fluorophores and even to 100 strains with three fluorophores as recently demonstrated using ribosomal frameshifting[21]. Such multiplexing capabilities

can be especially useful to characterize the input-output response of various candidate circuits (or the dependence of circuit behavior across a library of strain backgrounds) in parallel (using different light inductions across reactors). Immuno-beads can be used for more diverse cytometry-based measurements (the robot enabling automated incubation and wash, for example using the Opentrons OT-2 magnetic module). Technologies such as surface display[36,37] or GPCR signaling[38] can also be used to engineer biosensor strains to measure even more dimensions of the cultures with a single cytometer and at no reagent costs. Aside of high-performance quantitative strain characterization, such platforms can be useful for biotechnological applications[11]. Automated cytometry informing on the composition of artificial microbial consortia together with dynamic control of culture conditions (as demonstrated here using histidine auxotrophy and OD) could strongly reduce the need to engineer robust coexistence mechanisms[39], therefore enabling the use of a much larger diversity of consortia.

In the future, we hope that many ReacSight-based platforms will be assembled, and their design shared by a broad community to drastically expand our experimental capabilities, in order to address fundamental questions in microbiology and unlock the potential of synthetic biology in biotechnological applications.

## Methods

**Cloning and yeast strain construction.** All integrative plasmids are constructed using the modular cloning framework for yeast synthetic biology Yeast Tool Kit by Lee and colleagues[40] and all strains originate from the common laboratory strain BY4741. Strain genotypes are described in Supplementary Note 4. All strains used in this work express the light-inducible transcription factor EL222 from the *URA3* locus (transcriptional unit: pTDH3 NLS-VP16-EL222 tSSA1, common parental strain yIB32). Single-color constitutive expression strains (Fig. 2b, c) also harbor a pTDH3 FP tTDH1 transcriptional unit at the *LEU2* locus where FP is mCerulean, mNeon-Green or mScarlet-I. Corresponding CDS have been codon-optimized for expression in *S. cerevisiae*. The three-color strain harbors the same three transcriptional units in tandem (order: mCerulean, mNeonGreen, mScarlet-I) at the *LEU2* locus. The autofluorescence strain harbors an empty cassette at the *LEU2* locus to match auxotrophy markers between strains. For light-inducible gene expression (Figs. 2d and 3a, c), a pEL222 mNeonGreen tTDH1 transcriptional unit (where pEL222 is composed of 5 copies of the EL222 binding site followed by a truncated CYC1 promoter, originally named 5xBS-CYC180pr[17]) is integrated at the *LEU2* locus. To investigate the long-term stability of gene expression (Fig. 3d), three strains were built from the same parental strain (yIB90). In all cases, the optogenetic expression system is integrated in the *URA3* locus as explained above and a secretion stress reporter (pUPR mScarlet-I tENO1) is integrated in the *LEU2* locus. A light-responsive construct is integrated in the *HO* locus. This construct is mNeonGreen fused to three copies of the FLAG tag peptide in C-terminal (pEL222 mNeonGreen-3xFLAG tTDH1), or the same protein with the alpha-prepro secretion signal fused in N-terminal (pEL222 alpha-prepro-mNeonGreen-3xFLAG tTDH1), or the latter transcription unit present in triplicate ([pEL222 alpha-prepro-mNeonGreen-3xFLAG tTDH1]$_{3x}$). For histidine competition experiments (Fig. 4b), the histidine mutant strain (yIB90, parental strain yIB32) expresses a pUPR mScarlet-I tENO1 transcriptional unit integrated at the *LEU2* locus to report on the UPR activation. Here the pUPR promoter consists in four copies of a consensus UPR element[41] followed by a truncated CYC1 promoter. The histidine wild-type strain was obtained from the mutant strain yIB90 by integrating two identical pTDH3 mCerulean tTDH1 transcriptional units in tandem at the *HO* locus with *HIS3* selection, thereby restoring histidine prototrophy and enabling fluorescent barcoding. For the two-strain consortium experiment (Fig. 4c), the slow-growth histidine prototroph strain was obtained by integrating three identical pEL222 alpha-prepro-scFv-4-4-20 tTDH1 (burdensome secretion of an anti-fluorescein single-chain antibody fragment[42]) transcriptional units in tandem at the *HO* locus (*HIS3* selection) into yIB90 and blue light was used to induce the slow-growth phenotype.

**Yeast cell culture conditions.** All experiments were performed in 30 mL culture volume bioreactors at 30 degrees and in turbidostat mode (OD 0.5, typically corresponding to $10^7$ cells/mL according to cytometry data) with synthetic complete medium (Formedium LoFlo yeast nitrogen base CYN6510 and Formedium complete supplement mixture DCS0019) except for low histidine medium experiment where histidine drop-out amino-acid mixture was used (Sigma Y1751 or Formedium DCS0079) and complemented with desired levels of histidine (Sigma 53319).

**Cytometry acquisition and raw data analysis.** Gain settings of our cytometer (Guava EasyCyte 14HT, Luminex, with GuavaSoft v3.3) for all channels were set once and for all prior to the study such that yeast autofluorescence under our typical growth conditions is detectable but at the lower end of the 5-decade range of the instrument. We verified that cytometry data were reproducible week-to-week with these fixed settings. Single-color strains described above were used together with the autofluorescence control strain to obtain 'spectral' signatures of the three fluorophores *mCerulean*, *mNeonGreen*, and *mScarlet-I*, and autofluorescence levels for each channel. These signatures were also highly reproducible week-to-week (Supplementary Fig. 7). To convert raw cytometry data into fluorophore concentrations in relative promoter units (*RPU*[43]), we used a pipeline described in Supplementary Note 2. In essence, it uses data from single-color strains with pTDH3-driven expression for normalization. This pipeline was implemented in *Python*, mainly using NumPy[44] functions. The Opentrons OT-2 pipetting robot is used to wash the wells of the cytometer plate between samples, and dilute and load samples. We used the OT-2 Python API versions 1 and 2 (see https://opentrons.com).

**Model-predictive control.** For real-time control of gene expression using light (Fig. 3a, c), model-predictive control using the two-variables, three-parameters ODE model described in Fig. 3a was used. For state estimation upon arrival of cytometry data, the *FP* estimate was set equal to the fluorescence measurement (median of gated, deconvolved data) and the *mRNA* estimate was simply an 'open-loop' estimate based on simulating the history of light induction. This first state estimate corresponds to the state of the system at the time of sampling. To account for the time interval (and the concomitant light induction profile) between the sampling time and the data arrival time (typically 10–15 min), the model was used to obtain the corresponding updated state estimate. Then, a multi-dimensional, bounded, gradient-based search using SciPy[45] was used to find the best set of next light duty cycles minimizing the model-predicted distance to the target value over an horizon of 5 h (10 duty cycles).

**Histidine competition assays.** Precultures were performed in synthetic complete medium. Cells were washed in the same low histidine medium as the one used for turbidostat feeding of the competition culture and mixed with an approximate ratio mutant:WT of 5:1 (to ensure good statistics for long enough even when the mutant fitness is very low) before inoculation. Cytometry was acquired automatically every 2 h. At steady-state, the ratio between two competitors in a co-culture evolves exponentially at a rate equals to their growth rate difference. Linearity of the ratio logarithm for at least three time points was therefore used to assess when steady-state is reached. A threshold of 1 *mCerulean RPU* was used to assign each cell to its genotype. Size gating was performed as described in Supplementary Fig. 13 (parameters: size threshold = 0.5 and doublet threshold = 0.5, less stringent than for experiments of Figs. 2 and 3) to discard dead or dying cells.

**Dynamic control of the two-strain consortium.** For the original attempt presented in Fig. 4, a simple sigmoidal model describing the steady-state growth rate difference between the two strains as a function of OD was fitted on previous characterization data corresponding to different OD setpoints. Every 2 h, cytometry data were automatically acquired. To assign a genotype to each cytometry event, the GRN-B and ORG-G channels were used in combination, exploiting the fact that the histidine auxotroph strain is GRN-B positive and ORG-G negative (Supplementary Fig. 15). Based on the resulting estimate of the two-strain ratio, the model was used to optimize a vector of future OD setpoints (changing every 2 h for the next 10 h) using SciPy[45]. Additional results regarding characterization, strain identification, and ratio control are described in Supplementary Figs. 14–17.

**Bacterial strain and preculture conditions.** We used an E. coli strain from the natural isolates with low subcultures (NILS) collection[46]. The chosen strain, NILS18, has been isolated from a blood sample. It has a MIC of 2 mg/L of cefotaxime. Precultures, cultures, and experiments were performed in M9 liquid medium with 0.1% glucose, unless stated otherwise. For overnights, a single bacterial colony was picked from an agar plate and incubated overnight at 37 °C and 200 rpm. For precultures, the overnight was diluted to 0.05 $OD_{600}$ before a second incubation of 3–4 h under the same conditions, aiming to catch cells in exponential phase for the beginning of the experiment.

**Plate-reader experiments and data analysis.** For experiments, cells from precultures were diluted to an $OD_{600}$ of 0.05. Wells were either filled with 99 μL of M9 containing the necessary antibiotic concentration and 1 μL of the cell suspension (Fig. 5b) or with 198 μL of M9 containing the necessary antibiotic concentration and 2 μL of the cell suspension (Fig. 5c). Cefotaxime was purchased from Sigma-Aldrich and dilutions were made taking into account the purity specified by the vendor. We used 96-well plates with transparent flat bottoms (Corning 3370). A Spark multimode plate-reader (Tecan) was used for cell incubation at 37 °C and for OD acquisitions. We used the software of the vendor (Tecan Spark Control, v2.2). The incubation periods lasted 300 s, during which there was a first round of linear shaking (10 s, amplitude 5 mm, 390 rpm frequency), then still incubation, OD measurement, then two rounds of still incubation (20 s) and linear shaking (10 s, amplitude 5 mm, 390 rpm frequency) and finally still incubation for the rest of this cycle. $OD_{600}$ measurements were carried out every 5 min. For long-lasting

experiments ("turbidostat mode", Fig. 5b), if the median of the measured ODs was higher than a chosen threshold (0.05 or 0.1), the cultures were diluted by adding 100 μl of media with the chosen antibiotic concentration (same as at beginning of experiment), mixing, and then taking out 100 μl minus an estimate of the evaporated volume since last dilution. We estimated the evaporation rate to be 10 μL/h. Dilutions were repeated 14 or 15 times for each line of the microwell plate, depending on available media volume and tips for the pipetting robot. The robot was also instructed to add 100 μl of media with the chosen antibiotic concentration to the cultures if 10 h have elapsed since the beginning of the experiment with no dilution. Indeed, we estimated that the volume left after evaporation should then be very low. For repeated treatment experiments, 50 μL of antibiotic solution diluted in M9 media or of media only were added at the prescribed OD using the robot. All media were kept at 37 °C thanks to a Peltier module (Opentrons). The OD of the cell culture is estimated by subtracting from the measured $OD_{600}$ the $OD_{600}$ of a control well filled with M9 without bacteria. A few measurements presented aberrant values ($OD_{600} > 3 \ 10^{-3}$) during the first 2 h of the experiment and have been removed.

**Statistics and reproducibility**. No statistical method was used to predetermine sample size. No data were excluded from the analyses, excepted a few measurements during the first 2 h of the experiment in Fig. 5c that presented aberrant values ($OD_{600} > 3 \ 10^{-3}$). The experiments were not randomized. The Investigators were not blinded to allocation during experiments and outcome assessment.

**Reporting summary**. Further information on research design is available in the Nature Research Reporting Summary linked to this article.

## Data availability
All the raw experimental data generated in this study have been deposited on Zenodo (https://doi.org/10.5281/zenodo.4776009). Sequences of plasmids used to construct all yeast strains are available in the ReacSight Git repository (https://gitlab.inria.fr/InBio/Public/reacsight) in the GenBank format. Source data are provided with this paper.

## Code availability
In the ReacSight Git repository (https://gitlab.inria.fr/InBio/Public/reacsight), we provide (i) software supporting the ReacSight strategy (a full platform software template and examples of instrument APIs with corresponding Flask interfaces), (ii) code enabling automated gating and deconvolution of cytometry data and code enabling model-predictive control of gene expression using light, (iii) scripts performing data processing, analysis and plotting for figures, and (iv) a movie showing how to use ReacSight to set up experimental platforms (Supplementary Movie 1).

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

## Acknowledgements

The authors would like to thank Cosmin Saveanu, Emmanuel Frachon, and Alain Jacquier for the gift of the 16-vessel custom bioreactor setup and for guidance regarding its functioning, enabling us to customize and modernize it even further. We also received guidance and help from Albane Imbert (heading the Institut Pasteur *Fab Lab*) for the custom design and fabrication of Plexiglas and 3D-printed pieces. This work was supported by ANR grants CyberCircuits (ANR-18-CE91-0002; S.S.-C.), MEMIP (ANR-16-CE33-0018, S.S.-C.), Inception (ANR-16-CONV-0005, V.G.), and Anoruti (ANR-20-PAMR-0001, V.G.), by the H2020 Fet-Open COSY-BIO grant (grant agreement no. 766840, C.A., S.S.-C.) and by the Inria IPL grant COSY (G.B.).

## Author contributions

F.B., S.S.-C., and G.B. conceived the study. F.B. performed software and hardware engineering, performed experiments, analyzed data, and developed mathematical models. S.S.-C. developed strains with help of M.F, performed experiments with help of C.A., and analyzed data. V.G. helped with software and hardware developments, performed experiments, and analyzed data. A.F. helped with software and hardware developments. F.B. and G.B. supervised the study. F.B., S.S.-C., and G.B. wrote the manuscript with input from all authors.

## Competing interests

The authors declare no competing interests.
