## [Peer Review File · Nature Communications]

Reviewers' Comments:

Reviewer #1:

Remarks to the Author:

In the manuscript titled "Enhancing bioreactor arrays for automated measurements and reactive control with ReacSight", Bertaux et al present a DIY strategy for algorithmic, closed-loop data-responsive control ("reactive control", as the authors refer to it) of continuous cultures that can implement to a number of wide use low-cost / high-throughput bioreactors, liquid handling robots, and microwell-based analysis instruments. As a proof-of-concept, the authors developed an automated flow cytometry module that enables reactive control of growth conditions based on measured fluorescent protein reporter levels. In an impressive demonstration of the utility of the system, the authors conducted three case studies: 1) real time and tunable optogenetic control of gene expression, 2) using fluorescent barcodes to study the impact of nutrient scarcity on cell fitness and cell stress, and 3) maintaining ratios of two strains with differential fitness levels in a mixed culture.

I would argue that ReacSight is of interest and broadly useful to the systems, microbiology, and synthetic biology communities. It stands as an excellent entry into the accumulating collection of reported DIY/open source experimental platforms and could serve as a powerful extension of reported continuous growth systems (e.g., eVOLVER). In support of the author's claims, I am convinced it would be straightforward to implement their system given the clear description they provide in manuscript, and I could envision this platform enabling a diverse array of applications in experimental evolution, metabolic engineering, or the study of microbial consortia. Furthermore, the implementation appears to be eminently affordable. While generalizability and accessibility are two key criteria for success when developing a DIY/open source experimental platform, I would also argue that such platforms must be held to a high standard of robustness and reproducibility. The authors do a thorough job of validating the robustness of the individual components in their system (bioreactor arrays, yeast strains, liquid handler, and cytometer), but I think the manuscript would benefit from a more conclusive demonstration of the robustness of ReacSight as a system (see Major Comment #1). Secondly, in a final and critical experimental demonstration in figure 4, the authors maintained distinct ratios of two strains in a microbial consortium. However, the differences between the setpoints and the observed values are reported without a discussion (see Major Comment #2). After these issues are addressed, I believe the manuscript is suitable for publication in Nature Communications.

Major comments:

1. Demonstrating robustness and reproducibility of the system, not just the components. The authors demonstrated that each component of their system was reproducible, as shown in Figure 2 and Supplemental Figures 6, 7, and 8. However, in each of these cases a static protocol was used instead an actual experiment. Figure 3C shows the first example of a reactive experiment where cytometry measurements are used to guide bioreactor conditions, and it appears to have been repeated only once. To demonstrate the robustness and reproducibility their system, I suggest the authors show replicate runs of the experiments in 3C. Given that they have 8 available bioreactors, performing a pair of setpoints in triplicate, or performing the runs consecutively, would demonstrate that the system is able to dynamically respond to varying environmental factors and also generate reproducible results of gene expression at the various setpoints. Such a demonstration and a well-crafted discussion could help foster adoption of ReacSight and would be especially convincing for naive users who are skeptical that DIY components be used to construct robust laboratory systems (indeed they can with the help of good design principles and software!).

2. Address and discuss the results from Figure 4C. Inspection of plots in Figure 4C reveal that the setpoints do not match the observed ratios. Interestingly, the intended rank order is achieved, but ratios are off by a semi-consistent margin in most of the cultures. Unfortunately, the authors fail to adequately discuss these discrepancies other than a brief note in the figure caption that mentions "the presence of a slight steady-state error", but it is unclear what error they are referring to. The authors should fully discuss the results of Figure 4C in their manuscript, including an analysis of the potential failure mode leading to the discrepancy between set point and

observed results. The authors should also further clarify the steady-state error noted in the figure caption. As a suggestion, the authors could potentially strengthen their paper by iteratively refining their model (e.g., updating growth kinetics?), and then repeating the experiment. This would offer a powerful demonstration of ReacSight's customizability and ability to refine control algorithms to converge on a setup that achieves the desired setpoints.

Minor comments:

1. On line 165, the dynamic profiles can achieve "levels exceeding those obtained with the strong constitutive promoter pTDH3." There is no supporting data for this statement, or it is not clearly referenced.
2. Figure 2C, it is unclear if the single color and multi-color samples were run in parallel on the same day, or if they were run on separate days. Further, the OD plot is not presented for the multi-color reactor. This should be clarified in the writing. If the OD for the multi-color strain is not significantly different from single color, that can just be stated.
3. Experimental Controls are lacking in the experiments for Figure 4.
 - In Figure 4B, the authors co-culture 2 strains, and differentiate them based on the fact that the WT strain is expressing mCerulean and His- mutant is not. The authors should include a control experiment with a monoculture of the WT mCerulean-expressing cells to demonstrate that no loss of Cerulean expression occurs during culture growth over the same timescale as the experiment in 4B.
 - A similar control should be run for Figure 4C as well.
 - I highlight this as a minor comment instead of a major one because it is a comment on the experimental design, rather than on the ReacSight platform as presented. It should none-the-less be addressed.
4. Blue light is used in Figure 4C to trigger the slow growth phenotype of the slow His+ strain. However, the light intensity plot is not provided, and it is unclear if the light is applied constantly throughout the experiment. All of the other figures pulse the light and present light intensity plots. This should be clarified.
5. The authors briefly mention that other similar methods have not gained traction due to low throughput or cost. While throughput is addressed well, these cost differences are not well described. This might be a useful table to include in the supplement, outlining the various instruments and the costs at the time of publication.
6. The authors do not show doublet removal in their gating strategy for purposes of simplicity. A representative plot should still be included in the supplement to ensure reproducibility.
7. The authors show dilution data in the supplement using beads to validate re-using the same plate for the duration of the experiment. They appear to wash the plate with water. The supplementary figure is challenging to read, and it is difficult to draw conclusions as to the reliability of the wash. Is the timescale short enough that a little carry over will not bias the result?
8. Several figures (including figures 1, 2 and Supplementary Figures 1, 3, 7, and 8) are low resolution, making axis labels and legends hard to read. The authors should include high resolution images.
9. The authors normalize fluorescence obtained from flow cytometry by the FSC throughout the manuscript, to account for cell size. It is not common practice to normalize fluorescence by FSC in this manner, since forward scatter serves as an imprecise estimate for cell size at best. In fact, work such as Tzur et al (PLOS One, 2011) has suggested that cell volume normalization depends on the cell type and the cytometer, among other factors. I suggest the authors should either run controls showing that the normalization they have done accurately accounts for cell volume/size or cite supporting literature for normalizing in this manner (I am unaware of such demonstration). Alternatively, the overall fluorescence area can be reported directly, without accounting for cell

size since density-based cell gating and doublet discrimination should ensure that the majority of cells being analyzed are of roughly similar in size.

10. A video demonstration of the software and hardware interface/interaction provided in the supplement would strengthen the paper and facilitate understanding/impact.

11. Larger photos of the system, with clear callouts that don't obscure the components, would be useful.

12. The cut instructions for the acrylic should be made available as a more open-source file type (such as a .dxf file instead of an .ai file). I was able to load the file for the trash reservoir onto a 3D printer and open the cut instructions without errors. I was also able to read their code easily.

13. The authors have a line in their Opentrons code saying that "def get_log(): # does not seem to work?". As the detailed log file is an important part of the reproducibility of their work, this should be addressed and resolved before publication.

14. While figures are very clear overall, there are several examples of poor data visualization choices, including a rainbow color scheme in Fig 3C, an unlabeled axis in Fig 3A (right), unlabeled nutrient container icons in Fig 3A, and overall a difficult color palette for color blind readers.

Typographical/attribution errors:

1. I am unsure what a "funnel thrash" is. The OT-2 code refers to a printed_trash module, so I assume that is a persistent typo. It is named "thrash" consistently, including in the online repository.

2. Line 37, "when drug selection pressures increases...". Typo: should be "pressure increases" or "pressures increase"

3. On line 86 and 112: be consistent with other figures, label A/B instead of left/right

4. Line 110, "ReacSight also provide a solution...". Typo: should be "provides"

5. Line 117: capitalize "Arduinos"

6. Line 176: remove the "s" from (left photos)

7. Line 227: remove the "." from RPU.hr-1

8. Line 312: reference 6 is unclear whether a truly automated flow cytometry sampling system (i.e., self-loading samples) is in place

9. Line 368: Replace cf with "see"

10. Line 381: Add "Supplementary" to Figure 7A

11. Line 384: capitalize "Python"

12. Line 404: Add an S to Text 2.2

13. Line 415: capitalize "Zenodo"

14. Supplemental Figure 4 and 7 both have red lines in the figure, likely from a spell checker

15. In S2.1.2: remove space in "Reac Sight"

16. All references are verified, with the potential exception of reference 6 (see above)

Reviewer #2:

Remarks to the Author:

The authors present ReacSight - a flexible strategy to complement bioreactor arrays with additional instruments to measure cell and culture characteristics with high precision and integrate automated feedback functions to adjust culture conditions achieving desired cellular phenotypes or consortia compositions.

As an example, they integrate 2 types of bioreactor arrays (custom made or Chi.Bio) with an Opentrons liquid handling robot to manage off-sampling and sample processing for direct automated analysis in a flow cytometer. The high-quality single cell data is then used to determine the current state of the cultures and automatically derive model-based instructions for the bioreactors to steer cultures to desired states.

The authors explain how they combine three types of instruments (bioreactors, liquid handling robot, flow cytometer) to realise automated culture feedback and hence expand the set of parameters bioreactors typically used for process control like temperature, culture density, pH, dissolved oxygen or glucose by fluorescent and cell size measurement on the single molecule level leveraging a flow cytometer. They use this platform to illustrate applications previously shown for automated turbidostat setups like optogenetic control of gene expression, profiling consortia composition and set the stage for feedback-controlled experiments of more complex consortia.

Positives

1) The authors identify the need for smarter bioreactor arrays that allow for more properties to be measured continuously beyond classic bioreactor parameters. Such an enhancement will expand the data types and quality collected during the culture process and upon feedback integration also allow fine tuning of cellular and consortia states in continuous culture experiments.

2) They also emphasize the advantage of single cell data over bulk measurements and that expensive instruments needed for single cell measurement cannot be integrated into each bioreactor therefore need to be shared between them.

3) The authors present a credible scheme for how a range of instruments can be combined using a Python framework illustrating the integration via APIs and clicking based control of GUI-only software, supporting the generic applicability of ReacSight.

4) Three applications are illustrated – realising previously described turbidostat process control examples with automated continuous culture control over 8-40h based on cytometry data. The results suggest meaningful control of continuous cultures with good reproducibility can be achieved supporting the ReacSight functions for the yeast system used in this study.

Negatives

1) The publication of the integrated Chi.Bio bioreactor systems in July 2020 (Steel et al. 2020, referenced by the authors) already established a strong improvement of the state-of-the-art in that these bioreactors are low-cost and complete with a range of functionalities to measure and modulate cell culture properties (range of fluorescent reporters, LEDs for optogenetics, UV for mutation and white light) with pump and controller system that was shown to realise 2 out of 3 experiments used in this manuscript to highlight novel function enabled by ReacSight. The main improvement in this study is the addition of cytometry data to inform process control.

2) While the Opentrons liquid handler is affordable and can be easily integrated via Python, it's not clear it's a good choice given the reported issues around manual calibration requirements before each run and general reliability in long-term experiments.

3) The authors show that measurement and feedback can be applied every 20 minutes and most experiments are rather short (8-16h) and could potentially be performed in a normal workday. The final experiment running for 40h illustrates the need for an automated approach better.

4) The provided example experiments do not show a new capability or solve a specific problem, rather they seem to be a building block for future more complex experiments.

5) The arguments for affordability, accessibility and possible future expansion with additional instruments seem less than straight forward. True, the programming framework scales nicely and the API approach will generally work well. But the need to have a flow cytometer available to dedicate to such a platform seems a strong limiting factor for labs that would consider implementing ReacSight. Typically, these instruments are found in flow facilities rather than in labs since they are costly. The authors do not offer which additional instruments would be worth adding to such a platform to increase functionality – potentially this is quite limited to cytometry for the moment?

6) While the Chi.Bio integrated bioreactor solution was published only recently in Plos Biology and probably long after the initial work based on custom bioreactor arrays was under way – one could potentially consolidate and simplify the proposed framework by a cytometry addition to the Chi.Bio reactor platform within the Chi.Bio process control framework?

Additional comments

1) What are the limitations of the current implementation? How fast could samples be analysed and for how long can experiments be performed before random errors might stop the experiment?

2) Which type of organisms could be used in addition to the yeast system showcased – could one use fast-growing vibrio natriegens for instance?

3) It's not quite clear why an Opentrons liquid handler is needed – potentially a simple robotic arm (low-cost example: <https://automata.tech/about-eva/>) would suffice to manage off-sample well locations and loading the cytometer?

4) As a general strategy to combine different lab instruments the SiLA2 standard (<https://sila-standard.com>) seems to offer a good framework that might be considered to make the ReacSight approach more accessible?

Conclusion

The manuscript is well written and the experiments are nicely analysed and presented. The novelty presented focuses on the integration of cytometry data into the automated process control of bioreactor arrays. Adding sensitive single cell level fluorescence and size data to bioreactor process control is a promising approach and the authors also explained a relatively accessible strategy to integrate several instruments for such applications with their ReacSight strategy.

While they study a simple case for a dynamic ratio control of 2 co-cultured strains, a similar experiment could potentially be performed on the existing and accessible Chi.Bio bioreactor platform without the need for ReacSight (using 4 fluorescent channels rather than 2), which than would mean that none of the experiments presented could not have been performed on the existing Chi.Bio platform (Steel et al. 2020), which is also much more affordable than the proposed cytometry integration.

Reviewer #1 (Remarks to the Author):

In the manuscript titled “Enhancing bioreactor arrays for automated measurements and reactive control with ReacSight”, Bertaux et al present a DIY strategy for algorithmic, closed-loop data-responsive control (“reactive control”, as the authors refer to it) of continuous cultures that can implement to a number of wide use low-cost / high-throughput bioreactors, liquid handling robots, and microwell-based analysis instruments. As a proof-of-concept, the authors developed an automated flow cytometry module that enables reactive control of growth conditions based on measured fluorescent protein reporter levels. In an impressive demonstration of the utility of the system, the authors conducted three case studies: 1) real time and tunable optogenetic control of gene expression, 2) using fluorescent barcodes to study the impact of nutrient scarcity on cell fitness and cell stress, and 3) maintaining ratios of two strains with differential fitness levels in a mixed culture.

I would argue that ReacSight is of interest and broadly useful to the systems, microbiology, and synthetic biology communities. It stands as an excellent entry into the accumulating collection of reported DIY/open source experimental platforms and could serve as a powerful extension of reported continuous growth systems (e.g., eVOLVER). In support of the author’s claims, I am convinced it would be straightforward to implement their system given the clear description they provide in manuscript, and I could envision this platform enabling a diverse array of applications in experimental evolution, metabolic engineering, or the study of microbial consortia. Furthermore, the implementation appears to be eminently affordable. While generalizability and accessibility are two key criteria for success when developing a DIY/open source experimental platform, I would also argue that such platforms must be held to a high standard of robustness and reproducibility. The authors do a thorough job of validating the robustness of the individual components in their system (bioreactor arrays, yeast strains, liquid handler, and cytometer), but I think the manuscript would benefit from a more conclusive demonstration of the robustness of ReacSight as a system (see Major Comment #1). Secondly, in a final and critical experimental demonstration in figure 4, the authors maintained distinct ratios of two strains in a microbial consortium. However, the differences between the setpoints and the observed values are reported without a discussion (see Major Comment #2). After these issues are addressed, I believe the manuscript is suitable for publication in Nature Communications.

We thank the reviewer for recognizing the usefulness of ReacSight to a broad community. As detailed below, we believe we addressed all the issues raised.

Major comments

1. Demonstrating robustness and reproducibility of the system, not just the components. The authors demonstrated that each component of their system was reproducible, as shown in Figure 2 and Supplemental Figures 6, 7, and 8. However, in each of these cases a static protocol was used instead an actual experiment. Figure 3C shows the first example of a reactive experiment where cytometry measurements are used to guide bioreactor conditions, and it appears to have been repeated only once. To demonstrate the robustness and reproducibility their system, I suggest the authors show replicate runs of the experiments in 3C. Given that they have 8 available bioreactors, performing a pair of setpoints in triplicate, or performing the runs consecutively, would demonstrate that the system is able to dynamically respond to varying environmental factors and also generate reproducible results of gene expression at the various setpoints. Such a demonstration and a well-crafted discussion could help foster adoption of ReacSight and would be especially convincing for naive users who are skeptical that DIY components be used to construct robust laboratory systems (indeed they can with the help of good design principles and software!).

We agree that performing replicate runs of the experiment in Figure 3C will demonstrate the robustness and reproducibility of our system. Following the reviewer suggestion, we used all 8 reactors of the platform to perform, in parallel, two gene expression control experiments (selecting one low and one high expression target), each in quadruplicates. The results of this additional experiments are now shown in Supplementary Figure 10 and is also shown below. All replicates achieved excellent tracking of the target, and the light profiles decided by the control algorithms were highly similar, yet not identical, between replicates of the same target. Although the experiment was conducted several months later, the results were very similar to our original experiment, demonstrating the long-term reproducibility offered by the platform.

We added the following sentence in the main text to discuss these results (lines 208-212):

To further demonstrate the robustness and reproducibility of the platform, we performed several months later another 8-reactor experiment involving quadruplicate reactor runs for two different fluorophore level targets (Figure 10 in Supplementary Text S1). All replicates achieved excellent tracking of the target, and the light profiles decided by the control algorithms were highly similar, yet not identical, between replicates of the same target.

2. Address and discuss the results from Figure 4C. Inspection of plots in Figure 4C reveal that the setpoints do not match the observed ratios. Interestingly, the intended rank order is achieved, but ratios are off by a semi-consistent margin in most of the cultures. Unfortunately, the authors fail to adequately discuss these discrepancies other than a brief note in the figure caption that mentions “the presence of a slight steady-state error”, but it is unclear what error they are referring to. The authors should fully discuss the results of Figure 4C in their manuscript, including an analysis of the potential failure mode leading to the discrepancy between set point and observed results. The authors should also further clarify the steady-state error noted in the figure caption. As a suggestion, the authors could potentially strengthen their paper by iteratively refining their model (e.g., updating growth kinetics?), and then repeating the experiment. This would offer a powerful demonstration of ReacSight’s customizability and ability to refine control algorithms to converge on a setup that achieves the desired setpoints.

We agree with the reviewer. We have investigated the factors limiting the controllability of the two-strain ratio in our original experiments. We identified that an unexpected recovery of the growth rate of the slow strain was likely to play a key role. The specific composition of the medium might be promoting adaptation to the burdensome protein expression of the slow strain. Indeed, cell behaviors differ in subtle but significant ways when using media from different suppliers (Formedium and Sigma have different amino acid supplement mixtures). This led us to consider a new composition of our low histidine medium, much closer to our standard medium, that we used to characterize all our new strains. We then performed a new characterization of the behavior of the two strains in this new medium (Figure 12 in Supplementary Text 1). Using this new data, we improved the strain identification strategy (Figure 13 in Supplementary Text 1) and we updated the model parameterization of competition dynamics (Figure 14 in Supplementary Text 1). Finally, we combined these efforts to perform an additional real-time control experiment of the two-strain consortium in the new medium (Figure 15 in Supplementary Text 1, also shown below). Our controller now achieves to reach the target levels. However, we also observe significant oscillations around the targets.

These oscillations are likely caused by delays not accounted for in the model. Because functional bi-directional steering is now robustly achieved over more than 40 hours, we think these results are sufficient to illustrate the capabilities of ReacSight to greatly facilitate the development and control of engineered microbial consortia.

All these new results are now detailed in the new section 3.3 of Supplementary Text 1. We also provide directions for future improvements of the model and control strategy. In the main text and legend of Figure 4C, we now refer to the Supplementary Information for this in-depth investigation of the observed discrepancies. We also updated the Methods section accordingly.

Minor comments

1. On line 165, the dynamic profiles can achieve “levels exceeding those obtained with the strong constitutive promoter pTDH3.” There is no supporting data for this statement, or it is not clearly

referenced.

We now include in the Supplementary Information (Figure 9) a plot comparing in detail the data of Figure 2C and 2D regarding mNeonGreen expression, highlighting that optogenetic induction leads to expression levels slightly higher (around 10%) than pTDH3 after 4.5 hours of induction (timepoint #7).

2. Figure 2C, it is unclear if the single color and multi-color samples were run in parallel on the same day, or if they were run on separate days. Further, the OD plot is not presented for the multi-color reactor. This should be clarified in the writing. If the OD for the multi-color strain is not significantly different from single color, that can just be stated.

We clarified that the OD plot for the multi-color culture is similar to the single-color cultures. We kept the sentence at the end of the legend “All bioreactor experiments presented in this figure were performed in parallel, the same day, with the custom bioreactor platform version” as it encompasses both the constitutive expression turbidostat cultures (Figure 2C) and light-induced gene expression characterization (Figure 2D).

3. Experimental Controls are lacking in the experiments for Figure 4.

- In Figure 4B, the authors co-culture 2 strains, and differentiate them based on the fact that the WT strain is expressing mCerulean and His- mutant is not. The authors should include a control experiment with a monoculture of the WT mCerulean-expressing cells to demonstrate that no loss of Cerulean expression occurs during culture growth over the same timescale as the experiment in 4B.
- A similar control should be run for Figure 4C as well.

I highlight this as a minor comment instead of a major one because it is a comment on the experimental design, rather than on the ReacSight platform as presented. It should none-the-less be addressed.

We agree and we have run these control experiments, confirming the stability of mCerulean expression in the WT strain, and also providing an estimate of mis-attribution error of our algorithm in real conditions: around 1/1000 cell was attributed a WT phenotype in mutant-

only cultures and around 1/1000 cell was attributed a mutant phenotype in WT-only cultures. This means that the range of ratios between 0.01 to 100 is well suited for the reliable estimation of growth rate differences. All the corresponding data was in this suitable range with a comfortable margin. The corresponding data is now illustrated by two plots in Supplementary Information Figure 11, in the new section describing additional material regarding the three case studies.

For the OD-based consortium control experiment of Figure 4C, as discussed earlier, we have run extensive additional control and characterization experiments (Supplementary Information Figures 12 and 13).

4. Blue light is used in Figure 4C to trigger the slow growth phenotype of the slow His⁺ strain. However, the light intensity plot is not provided, and it is unclear if the light is applied constantly throughout the experiment. All of the other figures pulse the light and present light intensity plots. This should be clarified.

We clarified that blue light was applied at time 0 and remained constantly on to induce and maintain the slow growth phenotype via expression of burdensome protein secretion.

5. The authors briefly mention that other similar methods have not gained traction due to low throughput or cost. While throughput is addressed well, these cost differences are not well described. This might be a useful table to include in the supplement, outlining the various instruments and the costs at the time of publication.

Cost comparisons for such platforms with different designs and capabilities (number of parallel cultures, temporal resolution, sample treatment), which are made by assembling several components of various origins (custom-made, commercial, mixed), is very difficult and somewhat arbitrary given that equipment prices are not public.

We can still say that as of now, open-source, open-hardware components such as a Chi.Bio 8-bioreactors array (with 4 pumps per reactor and a controller board) costs US\$ 5895 from LabMaker. The cost of an Opentrons OT-2 pipetting robot with one multi-channel pipette is somewhat similar (US\$ 6800). We have added this information in the Supplementary Information within the adequate sections.

6. The authors do not show doublet removal in their gating strategy for purposes of simplicity. A representative plot should still be included in the supplement to ensure reproducibility.

We now include such a plot in Supplementary Information Figure 7B, also shown below.

7. The authors show dilution data in the supplement using beads to validate re-using the same plate for the duration of the experiment. They appear to wash the plate with water. The supplementary figure is challenging to read, and it is difficult to draw conclusions as to the reliability of the wash. Is the timescale short enough that a little carry over will not bias the result?

We have improved the clarity of the plot (Supplementary Information Figure 8) and expanded the discussion of the expected impact of carry-over, which is very low. Briefly, less than one out of 600-800 cells will be left in the acquisition plate after the wash, and these cells won't be able to grow in-between timepoints because of lack of medium.

8. Several figures (including figures 1, 2 and Supplementary Figures 1, 3, 7, and 8) are low resolution, making axis labels and legends hard to read. The authors should include high resolution images.

We thank the reviewer for noting this. We found out the issue is at the level of the word to pdf conversion. We corrected the issue.

9. The authors normalize fluorescence obtained from flow cytometry by the FSC throughout the manuscript, to account for cell size. It is not common practice to normalize fluorescence by FSC in this manner, since forward scatter serves as an imprecise estimate for cell size at best. In fact, work such as Tzur et al (PLOS One, 2011) has suggested that cell volume normalization depends on the cell type and the cytometer, among other factors. I suggest the authors should either run controls showing that the normalization they have done accurately accounts for cell volume/size or cite supporting literature for normalizing in this manner (I am unaware of such demonstration). Alternatively, the overall fluorescence area can be reported directly, without accounting for cell size since density-based cell gating and doublet discrimination should ensure that the majority of cells being analyzed are of roughly similar in size.

We have added a detailed discussion on the topic cell size normalization in the section "Metrology and automated analysis of cytometry data from yeast cultures" of the Supplementary Information. Briefly, we acknowledge that while several works use the forward scatter for cell size normalization, other works use the side scatter, and we also recognize that probably no perfect, instrument- and condition-independent proxy for cell size can be computed from cytometry-based optical measurements. We thus used a data-driven approach based on the reduction of the cell-to-cell variability for constitutive gene expression to justify the use of the forward scatter. We also compared the shape of the FSC and SSC distributions with published distributions of yeast cell size obtained with Coulter counters and found that the FSC distribution resembled it more.

10. A video demonstration of the software and hardware interface/interaction provided in the supplement would strengthen the paper and facilitate understanding/impact.

11. Larger photos of the system, with clear callouts that don't obscure the components, would be useful.

We have now edited a video highlighting key elements of a ReacSight platform and the key steps to run an experiment on a ReacSight platform. It also provides an additional view of the

different components for automating cytometry acquisition with the OT-2. The video is available in the ReacSight gitlab repository:

https://gitlab.inria.fr/InBio/Public/reacsight/-/blob/master/reacsight_presentation_video.mp4

12. The cut instructions for the acrylic should be made available as a more open-source file type (such as a .dxf file instead of an .ai file). I was able to load the file for the trash reservoir onto a 3D printer and open the cut instructions without errors. I was also able to read their code easily.

We have added files for cut instructions in open format (.svg and/or .dxf).

13. The authors have a line in their Opentrons code saying that “def get_log(): # does not seem to work?”. As the detailed log file is an important part of the reproducibility of their work, this should be addressed and resolved before publication.

We thank the reviewer for pointing this out. In fact, during testing we found that the *opentrons.robot.commands()* method of the Opentrons API v1 does not function as expected, preventing us to retrieve a list of elementary robot operations. However, because the instrument control does log all the commands sent to the OT2 Flask app as well as the responses of the Flask app, the platform will still generate a very detailed log because the robot actions exposed by the Flask app are very elementary. We have therefore removed the corresponding method in our Opentrons Flask app code.

14. While figures are very clear overall, there are several examples of poor data visualization choices, including a rainbow color scheme in Fig 3C, an unlabeled axis in Fig 3A (right), unlabeled nutrient container icons in Fig 3A, and overall a difficult color palette for color blind readers.

We thank the reviewer for pointing this out. We have changed the color scheme in Figure 3C. For Figure 3A, we chose not to repeat the Time label as it is indicated for the light intensity plot, but we added the ticks and tick labels to help the reader. We also did the same for Figure 3C, left.

Typographical/attribution errors

1. I am unsure what a “funnel thrash” is. The OT-2 code refers to a printed_trash module, so I assume that is a persistent typo. It is named “thrash” consistently, including in the online repository.

We now use the better term “waste funnel” consistently.

2. Line 37, “when drug selection pressures increases...”. Typo: should be “pressure increases” or “pressures increase”
3. On line 86 and 112: be consistent with other figures, label A/B instead of left/right
4. Line 110, “ReacSight also provide a solution...”. Typo: should be “provides”
5. Line 117: capitalize “Arduinos”
6. Line 176: remove the “s” from (left photos)
7. Line 227: remove the “.” from RPU.hr-1

2-7: We thank the reviewer for pointing these issues that we fixed.

8. Line 312: reference 6 is unclear whether a truly automated flow cytometry sampling system (i.e., self-loading samples) is in place

Indeed, sampling was manual in reference 6, we thank the reviewer for pointing this out and have adapted the manuscript accordingly.

8. Line 368: Replace cf with “see”
9. Line 381: Add “Supplementary” to Figure 7A
11. Line 384: capitalize “Python”
12. Line 404: Add an S to Text 2.2
13. Line 415: capitalize “Zenodo”
14. Supplemental Figure 4 and 7 both have red lines in the figure, likely from a spell checker
15. In S2.1.2: remove space in “Reac Sight”

9-15: We thank the reviewer for pointing these issues that we fixed.

16. All references are verified, with the potential exception of reference 6 (see above)

We are grateful to the reviewer for this very useful verification of our reference list. As mentioned above, we have updated how we cite reference 6.

Reviewer #2 (Remarks to the Author):

The authors present ReacSight - a flexible strategy to complement bioreactor arrays with additional instruments to measure cell and culture characteristics with high precision and integrate automated feedback functions to adjust culture conditions achieving desired cellular phenotypes or consortia compositions.

As an example, they integrate 2 types of bioreactor arrays (custom made or Chi.Bio) with an Opentrons liquid handling robot to manage off-sampling and sample processing for direct automated analysis in a flow cytometer. The high-quality single cell data is then used to determine the current state of the cultures and automatically derive model-based instructions for the bioreactors to steer cultures to desired states.

The authors explain how they combine three types of instruments (bioreactors, liquid handling robot, flow cytometer) to realise automated culture feedback and hence expand the set of parameters bioreactors typically used for process control like temperature, culture density, pH, dissolved oxygen or glucose by fluorescent and cell size measurement on the single molecule level leveraging a flow cytometer. They use this platform to illustrate applications previously shown for automated turbidostat setups like optogenetic control of gene expression, profiling consortia composition and set the stage for feedback-controlled experiments of more complex consortia.

Positives

1) The authors identify the need for smarter bioreactor arrays that allow for more properties to be measured continuously beyond classic bioreactor parameters. Such an enhancement will expand the data types and quality collected during the culture process and upon feedback integration also allow fine tuning of cellular and consortia states in continuous culture experiments.

2) They also emphasize the advantage of single cell data over bulk measurements and that expensive instruments needed for single cell measurement cannot be integrated into each bioreactor therefore need to be shared between them.

3) The authors present a credible scheme for how a range of instruments can be combined using a Python framework illustrating the integration via APIs and clicking based control of GUI-only software, supporting the generic applicability of ReacSight.

4) Three applications are illustrated – realising previously described turbidostat process control examples with automated continuous culture control over 8-40h based on cytometry data. The results suggest meaningful control of continuous cultures with good reproducibility can be achieved supporting the ReacSight functions for the yeast system used in this study.

We thank the reviewer for his positive assessment.

Negatives

1) The publication of the integrated Chi.Bio bioreactor systems in July 2020 (Steel et al. 2020, referenced by the authors) already established a strong improvement of the state-of-the-art in that these bioreactors are low-cost and complete with a range of functionalities to measure and modulate cell culture properties (range of fluorescent reporters, LEDs for optogenetics, UV for mutation and white light) with pump and controller system that was shown to realise 2 out of 3 experiments used in this manuscript to highlight novel function enabled by ReacSight. The main improvement in this study is the addition of cytometry data to inform process control.

The main purpose of ReacSight is to expand the capabilities of bioreactor arrays by providing an easy and generic strategy to automate the dynamic characterization of culture samples with very sensitive but bulky and rather expensive measurement devices such as cytometers.

More specifically, while it is true that the gene expression control case study and potentially (with some caveats) the two-strain control case study could have been performed using the Chi.Bio bioreactors alone, we do not see sampling automation for micro-well plate-based analysis instruments, most notably cytometers, as a “small” or “incremental” improvement, for several reasons.

First, because the fluorescence measurements are much more sensitive and specific, we obtain data of much higher quality. For instance, for light-controlled gene expression, we could build a highly predictive mathematical model of the gene expression response to various light inputs, enabling us to use a model-predictive control approach with great success. Also, there is no need to perform device-specific calibration because the same instrument is used for all bioreactors.

Second, in the context of leveraging optogenetics in genetic circuits, a great advantage of quantifying fluorescence of samples that are removed from the bioreactors is that the light excitations needed to measure fluorescence are not interfering with light-responsive proteins of the cell population in the bioreactor. This enables for example the simultaneous orthogonal control of multiple optogenetic systems (blue light and green/red or red/far-red for example) while still using blue and green fluorescent reporters.

Third, the use of a programmable pipetting robot in-between the bioreactors and the measurement device provides a unique opportunity to perform treatment operations on raw bioreactor samples before transfer to the measurement device. While we only used the robot to dilute the culture samples, more advanced measurement strategies can be foreseen, as already mentioned in the discussion of the main text.

Finally, and this is to us a very important point, the cytometer provides single-cell level data. It enables more informative characterization of an isogenic population (cell-to-cell

heterogeneity, bi-modal phenotypes, cell-cycle stages...) and it also opens up exciting multiplexing capabilities, where many different genotypes and their associated phenotypes can be resolved from co-cultures thanks to fluorescence barcoding strategies. Our second case study provides a first proof of concept around this idea, but as already stated in the discussion of the main text, we only touched the surface of the application potential of such platforms.

2) While the Opentrons liquid handler is affordable and can be easily integrated via Python, it's not clear it's a good choice given the reported issues around manual calibration requirements before each run and general reliability in long-term experiments.

It is not our experience that the OT-2 should be calibrated again before each run. In fact, over 2.5 years we calibrated it only twice. It might be in part because our OT-2 is solely used for this task, and it might also be related to the fact that when onboarding custom labware we calibrate its positional parameters only via the Python API and not the OT-2 GUI application, which in our hands can introduce some calibration problems. Also, to highlight the reliability of the OT-2, we did not encounter issues and did not need to do any kind of maintenance on the instrument over these 2.5 years.

3) The authors show that measurement and feedback can be applied every 20 minutes and most experiments are rather short (8-16h) and could potentially be performed in a normal workday. The final experiment running for 40h illustrates the need for an automated approach better.

We adapt the duration of experiments according to their context so as to optimize the use of the platform. In some cases, we ran reactive experiments for more than 100 hours (see plot below from Aditya et al., Nature Communications, 2021).

Also, it is important to note that the reported duration of the experiments do not account for the interval between inoculation and the beginning of the measurements, stimulation and potential reactive program. This interval is usually made of 14-18 hours so as to reach a steady physiology for the cell population. Finally, even for relatively short experiments, the amount of time freed thanks to automation and the benefits of having a standardized format for data generation and detailed data logs, improved reproducibility and integration with real-time data processing, are still very valuable.

4) The provided example experiments do not show a new capability or solve a specific problem, rather they seem to be a building block for future more complex experiments.

The purpose of our paper is to present the ReaSight strategy and convincingly demonstrate its potential with non-trivial case studies. In that sense, it is indeed a building block for future biological investigations. However, we think it is an important building block because of its enabling potential.

5) The arguments for affordability, accessibility and possible future expansion with additional instruments seem less than straight forward. True, the programming framework scales nicely and the API approach will generally work well. But the need to have a flow cytometer available to dedicate to such a platform seems a strong limiting factor for labs that would consider implementing ReaSight. Typically, these instruments are found in flow facilities rather than in labs since they are costly. The authors do not offer which additional instruments would be worth adding to such a platform to increase functionality – potentially this is quite limited to cytometry for the moment?

We agree that cytometers are most commonly found in shared facilities, however the price of benchtop cytometers such as our Guava EasyCyte (which starts at 60-70 k€) can be compatible for single-lab purchases, especially given that, thanks to measurement automation, their effective use time can be very high, meaning that the return on investment can be very good. Alternatively, plate-readers (around 20 k€) can provide bulk optical and fluorescence measurements. While the Chi.Bios can provide such measurements in-situ, using a plate-reader within a ReaSight platform will enable more diverse measurements (for example, colorimetric enzymatic assays in combination with sample treatment capabilities offered by the OT-2), and also avoids the problem of undesired optogenetic stimulation caused by fluorescence measurements.

6) While the Chi.Bio integrated bioreactor solution was published only recently in Plos Biology and probably long after the initial work based on custom bioreactor arrays was under way – one could potentially consolidate and simplify the proposed framework by a cytometry addition to the Chi.Bio reactor platform within the Chi.Bio process control framework?

The instrument control architecture of ReaSight, where a variety of instruments can be controlled by a single experiment control computer via network-exposed python APIs, has the advantage of offering modularity and versatility. It is true that experiment control could be in theory integrated with the Chi.Bio app, and because ReaSight software is open-source this can be done by users committed to using Chi.Bio bioreactors only. We prefer to further highlight and strengthen the versatility of ReaSight by working on use cases involving other instruments. For example, we are currently working on demonstrating how ReaSight can be used to drive the eVOLVER bioreactor system (Wong et al., Nature Biotechnology, 2018).

Additional comments

1) What are the limitations of the current implementation? How fast could samples be analysed and for how long can experiments be performed before random errors might stop the experiment?

The maximal temporal resolution (time between cytometry timepoints) that we can reach is influenced by several factors (time needed to wash the cytometer plate, time for the cytometry acquisition and the cytometer cleaning program). With eight reactors, 5000 events per sample and an OD of 0.5, we can typically perform cytometry measurements every 10-15 minutes, which is short in comparison to the yeast cell cycle. Regarding how long experiments can be run before un-explained or impossible-to-predict errors affect the experiments, for our

workhorse platform, thanks to continuous software and procedure improvements, we reached a high reliability, with no issues for 90% of our experiments (running for 2 to 5 days typically). When issues occur, they are most of the time due to human errors, and very few remain unexplained. We should stress however that this required some software and procedure tuning that is in-part specific to a given ReaSight platform instance, so ReaSight users building a new platform might not reach such reliability out-of-the-box. Still, we used our practical experience to incorporate generic reliability-improving design principles in the ReaSight software.

2) Which type of organisms could be used in addition to the yeast system showcased – could one use fast-growing vibrio natriegens for instance?

Yes, we think the ReaSight strategy and in-particular ReaSight implementations achieving automated cytometry can be used for other microbial systems. The bioreactor system itself and its operation should of course be compatible (pump rate, avoidance of biofilm formation). The temporal resolution might need to be further optimized depending on the applications.

3) It's not quite clear why an Opentrons liquid handler is needed – potentially a simple robotic arm (low-cost example: <https://automata.tech/about-eva/>) would suffice to manage off-sample well locations and loading the cytometer?

An advantage of using a pipetting robot between bioreactors and the cytometer is to enable sample treatment operations to take place. For instance, in our case the ability to dilute samples is critical, as too high cell density impedes performance of the cytometry acquisition. Also, the space footprint of a robotic arm such as Eva does not seem that lower than the one of an OT-2.

4) As a general strategy to combine different lab instruments the SiLA2 standard (<https://sila-standard.com>) seems to offer a good framework that might be considered to make the ReaSight approach more accessible?

We thank the reviewer for the reference to the SiLA2 standard, which we did not know about. Standardization is indeed a key goal for our community. It seems however that SiLA2 has not really been adopted by academic labs of our community, so as of now making ReaSight based on the SiLA2 standard is unlikely to provide real gains for the ReaSight target audience. We however will watch closely the evolution of SiLA2 and consider it for future ReaSight releases.

Conclusion

The manuscript is well written and the experiments are nicely analysed and presented. The novelty presented focuses on the integration of cytometry data into the automated process control of bioreactor arrays. Adding sensitive single cell level fluorescence and size data to bioreactor process control is a promising approach and the authors also explained a relatively accessible strategy to integrate several instruments for such applications with their ReaSight strategy.

We thank the reviewer for recognizing the quality and usefulness of our work.

While they study a simple case for a dynamic ratio control of 2 co-cultured strains, a similar experiment could potentially be performed on the existing and accessible Chi.Bio bioreactor platform without the need for ReaSight (using 4 fluorescent channels rather than 2), which than would mean that none of the experiments presented could not have been performed on the existing Chi.Bio platform (Steel et al. 2020), which is also much more affordable than the proposed cytometry integration.

As mentioned earlier, we agree that a standalone Chi.Bio bioreactor array has already powerful capabilities, but we also believe that the additional benefits of using ReacSight to automate ex-situ characterization of culture samples using a single high-throughput measurement device can be very substantial both in terms of data quality but also in terms of broadening the range of possible measurements and experiments. This is especially true when the measurement device is a cytometer, because it provides high sensitivity and single-cell resolution, which opens a broad range of innovative applications leveraging strain multiplexing in a single reactor (our second case study providing a simple proof of concept of such multiplexing). We want to note that this is also true for more affordable measurement devices such as plate readers. Using a single instrument to measure samples from all cultures removes the need for reactor-specific calibration. Also, the pipetting robot can be used to dilute samples, increasing the dynamic range of the measurements. More sophisticated sample treatment protocols with the OT-2 can also enable other types of measurements, such as enzymatic assays. Ex-situ measurements are also useful because optogenetic control (usually quite sensitive to light) and fluorescence measurements (that require significant light intensities) are not convenient to combine on the same cells. Finally, in-situ fluorescence measurements can be complex to interpret when the optical density of the culture also changes (although this effect was carefully investigated in the Chi.Bio publication by Steel and colleagues).

Reviewers' Comments:

Reviewer #1:

Remarks to the Author:

Overview

In their revision, the authors have done a great job of addressing my comments. The supplemental data they provide is clearly presented and greatly strengthens their argument that ReacSight is successful as an adaptable, flexible system that enables automated experimental sampling, testing, and reactive control. I am genuinely impressed that they were able to rapidly generate the data demonstrating robustness and reproducibility data of their platform, as well as thoroughly interrogate and then update their platform in response to Major Comments 1 and 2.

Detailed comments for each response are provided below. To summarize, I recommend that the authors address the following additional concerns prior to publication, which are further explained in response to their comments:

- Add a statement in the main text about their major conclusions about the ReacSight system that recapitulates their response to Major Comment 2.
- Review and edit the manuscript and supplement for clarity and grammar and, critically, ensure that references in the text to the supplement are consistent.

I also suggest a few visual design choices that may strengthen the paper that the authors can implement or disregard at their discretion. See Major Comment 2 and Minor Comment 14, below.

Response to Author's Responses:

Major Comment 1:

We agree that performing replicate runs of the experiment in Figure 3C will demonstrate the robustness and reproducibility of our system. Following the reviewer suggestion, we used all 8 reactors of the platform to perform, in parallel, two gene expression control experiments (selecting one low and one high expression target), each in quadruplicates. The results of this additional experiments are now shown in Supplementary Figure 10 and is also shown below. All replicates achieved excellent tracking of the target, and the light profiles decided by the control algorithms were highly similar, yet not identical, between replicates of the same target. Although the experiment was conducted several months later, the results were very similar to our original experiment, demonstrating the long-term reproducibility offered by the platform.

We added the following sentence in the main text to discuss these results (lines 208-212):

To further demonstrate the robustness and reproducibility of the platform, we performed several months later another 8-reactor experiment involving quadruplicate reactor runs for two different fluorophore level targets (Figure 10 in Supplementary Text S1). All replicates achieved excellent tracking of the target, and the light profiles decided by the control algorithms were highly similar, yet not identical, between replicates of the same target.

The additional data convincingly demonstrate that the system generates robust and reproducible results. The authors have resolved my concern. See Minor comment 14 below for further suggestions.

Major comment 2:

We agree with the reviewer. We have investigated the factors limiting the controllability of the two-strain ratio in our original experiments. We identified that an unexpected recovery of the growth rate of the slow strain was likely to play a key role. The specific composition of the medium might be promoting adaptation to the burdensome protein expression of the slow strain. Indeed, cell behaviors differ in subtle but significant ways when using media from different suppliers (Formedium and Sigma have different amino acid supplement mixtures). This led us to consider a new composition of our low histidine medium, much closer to our standard medium, that we used to characterize all our new strains. We then performed a new characterization of the behavior of the two strains in this new medium (Figure 12 in Supplementary Text 1). Using this new data, we improved the strain identification strategy (Figure 13 in Supplementary Text 1) and we updated the model parameterization of competition dynamics (Figure 14 in Supplementary Text 1). Finally, we combined these efforts to perform an additional real-time control experiment of the two-strain consortium in the new medium (Figure 15 in Supplementary Text 1, also shown below). Our controller now achieves to reach the target levels. However, we also observe significant oscillations around the targets. These oscillations are likely caused by delays not accounted for in the model. Because functional bi-directional steering is now robustly achieved over more than 40 hours, we think these results are sufficient to illustrate the capabilities of ReacSight to greatly facilitate the development and control of engineered microbial consortia. All these new results are now detailed in the new section 3.3 of Supplementary Text 1. We also provide directions for future improvements of the model and control strategy. In the main text and legend of Figure 4C, we now refer to the Supplementary Information for this in-depth investigation of the observed discrepancies. We also updated the Methods section accordingly.

The authors chose to revisit their experimental conditions, update their model, and conduct experimental replicates, a decision I applaud. The authors achieved the targeted ratios but identified new issues with their experimental conditions that produced oscillations due to delays in growth rate while applying reactive control. Given that the core focus of this paper is not about maintaining set ratios within a two-strain consortium, I believe the authors successfully demonstrate the ReacSight platform and its capability for reactive control, as well as the ability to rapidly modify their system and software for new control schemes and models.

I would ask the authors to make the following changes before publication:

- Add a sentence or two in the main text at line 307 stating their major conclusions about the ReacSight system/platform that summarizes the arguments presented in the supplement,
- The major revisions to the supplement resulted in several incorrect callouts in the main text. The following are a few examples—please carefully review and revise the manuscript for consistency.
 - Supplemental figures 12-15 are labeled as supplemental figures 7-10
 - Lines 272 and 307: I noticed a change in language in the manuscript, “Supplementary Text S1 (section 3.3)”
 - Line 191 and 196 of the manuscript refer to S1.2.1 for the custom array and S1.2.2 for the Chi.Bio reactors. By my interpretation, S1.2.1 refers to “Optogenetic-

enabled bioreactor set ups” and S1.2.2 refers to “Metrology and automated...” Should these callouts be S1.2.1.1 and S1.2.1.2? It appears that I missed this in the first revision, but it is much more apparent now that the supplement contains these critical pieces of data.

- Line 394 of the manuscript references S2.2, is that S1.2.2?
- Line 357: the strain table is now Table 1 of Supplemental Text S1.4

The following are suggestions that the authors to consider:

- Strain genotype information in Figure 4C appears to be inconsistent with data in the supplement. According to S1.4 Table 1, the histidine auxotroph strain yIB88 expresses mNeonGreen, and the slow growth strain with burdensome protein expression yIB135 expresses mScarlet. I would suggest the authors switch the orange and green color for the cartoon yeasts throughout Figure 4C.

Minor Comment 1:

We now include in the Supplementary Information (Figure 9) a plot comparing in detail the data of Figure 2C and 2D regarding mNeonGreen expression, highlighting that optogenetic induction leads to expression levels slightly higher (around 10%) than pTDH3 after 4.5 hours of induction (timepoint #7).

The addition made by the authors has addressed my concern.

Minor Comment 2:

We clarified that the OD plot for the multi-color culture is similar to the single-color cultures. We kept the sentence at the end of the legend “All bioreactor experiments presented in this figure were performed in parallel, the same day, with the custom bioreactor platform version” as it encompasses both the constitutive expression turbidostat cultures (Figure 2C) and light induced gene expression characterization (Figure 2D).

The authors explanation has addressed my concern

Minor Comment 3:

We agree and we have run these control experiments, confirming the stability of mCerulean expression in the WT strain, and also providing an estimate of mis-attribution error of our algorithm in real conditions: around 1/1000 cell was attributed a WT phenotype in mutantonly cultures and around 1/1000 cell was attributed a mutant phenotype in WT-only cultures. This means that the range of ratios between 0.01 to 100 is well suited for the reliable estimation of growth rate differences. All the corresponding data was in this suitable range with a comfortable margin. The corresponding data is now illustrated by two plots in Supplementary Information Figure 11, in the new section describing additional material regarding the three case studies.

For the OD-based consortium control experiment of Figure 4C, as discussed earlier, we have run extensive additional control and characterization experiments (Supplementary

Information Figures 12 and 13).

The addition described by the authors has addressed this comment

Minor Comment 4:

We clarified that blue light was applied at time 0 and remained constantly on to induce and maintain the slow growth phenotype via expression of burdensome protein secretion.

The authors explanation has addressed this concern

Minor Comment 5:

Cost comparisons for such platforms with different designs and capabilities (number of parallel cultures, temporal resolution, sample treatment), which are made by assembling several components of various origins (custom-made, commercial, mixed), is very difficult and somewhat arbitrary given that equipment prices are not public. We can still say that as of now, open-source, open-hardware components such as a Chi.Bio 8-bioreactors array (with 4 pumps per reactor and a controller board) costs US\$ 5895 from LabMaker. The cost of an Opentrons OT-2 pipetting robot with one multi-channel pipette is somewhat similar (US\$ 6800). We have added this information in the Supplementary Information within the adequate sections.

The authors have adequately addressed the concern.

Minor Comment 6:

We now include such a plot in Supplementary Information Figure 7B, also shown below.

The addition described by the authors has addressed this comment

Minor Comment 7:

We have improved the clarity of the plot (Supplementary Information Figure 8) and expanded the discussion of the expected impact of carry-over, which is very low. Briefly, less than one out of 600-800 cells will be left in the acquisition plate after the wash, and these cells won't be able to grow in-between timepoints because of lack of medium.

The addition described by the authors has addressed this comment

Minor Comment 8:

We thank the reviewer for noting this. We found out the issue is at the level of the word to pdf conversion. We corrected the issue.

The author's explanation has addressed this concern

Minor Comment 9:

We have added a detailed discussion on the topic cell size normalization in the section “Metrology and automated analysis of cytometry data from yeast cultures” of the Supplementary Information. Briefly, we acknowledge that while several works use the forward scatter for cell size normalization, other works use the side scatter, and we also recognize that probably no perfect, instrument- and condition-independent proxy for cell size can be computed from cytometry-based optical measurements. We thus used a data-driven approach based on the reduction of the cell-to-cell variability for constitutive gene expression to justify the use of the forward scatter. We also compared the shape of the FSC and SSC distributions with published distributions of yeast cell size obtained with Coulter counters and found that the FSC distribution resembled it more.

I appreciate the authors providing literature for both sides of the argument and clearly justifying their choice.

Minor Comment 10-11:

We have now edited a video highlighting key elements of a ReacSight platform and the key steps to run an experiment on a ReacSight platform. It also provides an additional view of the different components for automating cytometry acquisition with the OT-2. The video is available in the ReacSight gitlab repository: https://gitlab.inria.fr/InBio/Public/reacsight/-/blob/master/reacsight_presentation_video.mp4

I appreciate the additional effort to highlight the system with a video. I was unable to follow the link above but was able to find the video through the repository.

Minor Comment 12:

We have added files for cut instructions in open format (.svg and/or .dxf).

The addition described by the authors has addressed this comment

Minor Comment 13:

We thank the reviewer for pointing this out. In fact, during testing we found that the `opentrons.robot.commands()` method of the Opentrons API v1 does not function as expected, preventing us to retrieve a list of elementary robot operations. However, because the instrument control does log all the commands sent to the OT2 Flask app as well as the responses of the Flask app, the platform will still generate a very detailed log because the robot actions exposed by the Flask app are very elementary. We have therefore removed the corresponding method in our Opentrons Flask app code.

The author's explanation has addressed this concern

Minor Comment 14:

We thank the reviewer for pointing this out. We have changed the color scheme in Figure 3C. For Figure 3A, we chose not to repeat the Time label as it is indicated for the light intensity plot, but we added the ticks and tick labels to help the reader. We also did the same for Figure 3C, left.

I would like to thank the authors for addressing my concerns, however I have a few suggestions for consideration:

- In Figure 3C right, changing the color palette from rainbow made it less distracting. However, the color gradient is being used to visually encode time, but the x axis being timepoints does that independently without color. I suggest that each panel on the right be the same color as the corresponding median values represented in the left panel. For example, change the target = 0.2 panel to dark blue for all timepoints, and the target =0.8 panel to lightest blue for all timepoints.
- The authors could also consider replacing the right panel of Figure 3C with the new data presented in the supplement in response to Major Comment 1 (currently Supplementary Figure 10), and move the “gene expression in single cells” panel to the supplement. This would allow them to highlight their excellent data showing that “real time control of gene expression using light” was robust and reproducible. An argument could be made either way on the utility of the two pieces of data (single cell vs. the new data), so I would leave it to the authors to decide what they think is best!

Typos/attribution errors:

We now use the better term “waste funnel” consistently.

I appreciate the broad revision, however “thrash” still appears in Supplemental Figure 5.

2-7: We thank the reviewer for pointing these issues that we fixed. Indeed, sampling was manual in reference 6, we thank the reviewer for pointing this out and have adapted the manuscript accordingly.

9-15: We thank the reviewer for pointing these issues that we fixed. We are grateful to the reviewer for this very useful verification of our reference list. As mentioned above, we have updated how we cite reference 6.

The authors address our specific concerns, but we would encourage them to do another detailed verification for typos throughout the manuscript and supplement as they carefully review and revise the manuscripts for consistency from Major Comment 2. The following are non-exhaustive examples:

- Line 115: capitalize “Python”
- Line 185-186: “is” changed to “are” 2x
- Line 380, 381, 383: Formedium is inconsistently capitalized

Reviewer #2:

Remarks to the Author:

The authors have addressed some of my concerns but from my perspective the study as presently described is more incremental than a substantial improvement on what has already been published.

To make the case for a substantial improvement the authors need to either show an integration of multiple devices or an exemplar application.

For example to show that ReacSight is a powerful generalisable framework to feedback from ex-situ measurements the authors need to demonstrate:

- (1) Multiple bioreactor systems can work with it (that's already shown in the ms)
- (2) Multiple external measurement devices can be managed- e.g. at least a plate reader (or alternative instrument) application
- (3) Liquid handling robot can do more useful things than simple 20x dilution (which could be done without a robot) -> they argue enzymatic assays might be useful but so not show any data to support this.

Alternatively the authors could show a novel application that is enabled thanks to ex-situ measurement:

- (1) e.g. 5 day experiments as shown by Aditya et al., Nature Communications 2021 - as a difficult thing to do but can be achieved by the authors developments
- (2) the argument that pulsed (short) in-situ fluorescence measurements would indeed cause major accidental optogenetic actuation and therefore require ex-situ analysis would have to be supported by some data
- (3) the multiplexing/barcoding of mixed populations they suggest might be possible with 10s of channels should be attempted with more ambition (rather than just 2 populations)
- (4) perhaps offer a novel application

We thank both reviewers for their detailed analysis of our work and for their suggestions to further improve our article.

Below we provide detailed comments to their remarks. Reviewers' comments are in black, with their citations of our responses to their first round of comments in light blue, and our responses are in dark blue.

Reviewer #1 (Remarks to the Author):

Overview

In their revision, the authors have done a great job of addressing my comments. The supplemental data they provide is clearly presented and greatly strengthens their argument that ReacSight is successful as an adaptable, flexible system that enables automated experimental sampling, testing, and reactive control. I am genuinely impressed that they were able to rapidly generate the data demonstrating robustness and reproducibility data of their platform, as well as thoroughly interrogate and then update their platform in response to Major Comments 1 and 2.

We thank the reviewer for their positive assessment of our work to revise and improve the paper.

Detailed comments for each response are provided below. To summarize, I recommend that the authors address the following additional concerns prior to publication, which are further explained in response to their comments:

- Add a statement in the main text about their major conclusions about the ReacSight system that recapitulates their response to Major Comment 2.
- Review and edit the manuscript and supplement for clarity and grammar and, critically, ensure that references in the text to the supplement are consistent.

I also suggest a few visual design choices that may strengthen the paper that the authors can implement or disregard at their discretion. See Major Comment 2 and Minor Comment 14, below.

We provide detailed answers to these comments below.

Response to Author's Responses

Major Comment 1:

We agree that performing replicate runs of the experiment in Figure 3C will demonstrate the robustness and reproducibility of our system. Following the reviewer suggestion, we used all 8 reactors of the platform to perform, in parallel, two gene expression control experiments (selecting one low and one high expression target), each in quadruplicates. The results of this additional experiments are now shown in Supplementary Figure 10 and is also shown below. All replicates achieved excellent tracking of the target, and the light profiles decided by the control algorithms were highly similar, yet not identical, between replicates of the same target. Although the experiment was conducted several months later, the results were very similar to our original experiment, demonstrating the long-term reproducibility offered by the platform.

We added the following sentence in the main text to discuss these results (lines 208-212):

To further demonstrate the robustness and reproducibility of the platform, we performed several months later another 8-reactor experiment involving quadruplicate reactor runs for two different fluorophore level targets (Figure 10 in Supplementary Text S1). All replicates achieved excellent tracking of the target, and the light profiles

decided by the control algorithms were highly similar, yet not identical, between replicates of the same target.

The additional data convincingly demonstrate that the system generates robust and reproducible results. The authors have resolved my concern. See Minor comment 14 below for further suggestions.

Thank you for having suggested this interesting addition.

Major Comment 2:

We agree with the reviewer. We have investigated the factors limiting the controllability of the two-strain ratio in our original experiments. We identified that an unexpected recovery of the growth rate of the slow strain was likely to play a key role. The specific composition of the medium might be promoting adaptation to the burdensome protein expression of the slow strain. Indeed, cell behaviors differ in subtle but significant ways when using media from different suppliers (Formedium and Sigma have different amino acid supplement mixtures). This led us to consider a new composition of our low histidine medium, much closer to our standard medium, that we used to characterize all our new strains. We then performed a new characterization of the behavior of the two strains in this new medium (Figure 12 in Supplementary Text 1). Using this new data, we improved the strain identification strategy (Figure 13 in Supplementary Text 1) and we updated the model parameterization of competition dynamics (Figure 14 in Supplementary Text 1). Finally, we combined these efforts to perform an additional real-time control experiment of the two-strain consortium in the new medium (Figure 15 in Supplementary Text 1, also shown below). Our controller now achieves to reach the target levels. However, we also observe significant oscillations around the targets. These oscillations are likely caused by delays not accounted for in the model. Because functional bi-directional steering is now robustly achieved over more than 40 hours, we think these results are sufficient to illustrate the capabilities of ReaSight to greatly facilitate the development and control of engineered microbial consortia. All these new results are now detailed in the new section 3.3 of Supplementary Text 1. We also provide directions for future improvements of the model and control strategy. In the main text and legend of Figure 4C, we now refer to the Supplementary Information for this in-depth investigation of the observed discrepancies. We also updated the Methods section accordingly.

The authors chose to revisit their experimental conditions, update their model, and conduct experimental replicates, a decision I applaud.

Thank you!

The authors achieved the targeted ratios but identified new issues with their experimental conditions that produced oscillations due to delays in growth rate while applying reactive control. Given that the core focus of this paper is not about maintaining set ratios within a two-strain consortium, I believe the authors successfully demonstrate the ReaSight platform and its capability for reactive control, as well as the ability to rapidly modify their system and software for new control schemes and models.

I would ask the authors to make the following changes before publication:

- Add a sentence or two in the main text at line 307 stating their major conclusions about the ReaSight system/platform that summarizes the arguments presented in the supplement,

This has been done. See lines 345 – 352.

- The major revisions to the supplement resulted in several incorrect callouts in the main text. The following are a few examples—please carefully review and revise the manuscript for consistency.
 - Supplemental figures 12-15 are labeled as supplemental figures 7-10
 - Lines 272 and 307: I noticed a change in language in the manuscript, “Supplementary Text S1 (section 3.3)”
 - Line 191 and 196 of the manuscript refer to S1.2.1 for the custom array and S1.2.2 for the Chi.Bio reactors. By my interpretation, S1.2.1 refers to “Optogenetic-enabled bioreactor set ups” and S1.2.2 refers to “Metrology and automated...” Should these callouts be S1.2.1.1 and S1.2.1.2? It appears that I missed this in the first revision, but it is much more apparent now that the supplement contains these critical pieces of data.
 - Line 394 of the manuscript references S2.2, is that S1.2.2?
 - Line 357: the strain table is now Table 1 of Supplemental Text S1.4

We have significantly revised and restructured the supplementary material. Now, it is organized in four Supplementary Notes. In the main text, we either make reference to Supplementary Notes or to specific Supplementary Figures. We have carefully checked all references. The above-mentioned issues have been solved.

The following are suggestions that the authors to consider:

Strain genotype information in Figure 4C appears to be inconsistent with data in the supplement. According to S1.4 Table 1, the histidine auxotroph strain γ B88 expresses mNeonGreen, and the slow growth strain with burdensome protein expression γ B135 expresses mScarlet. I would suggest the authors switch the orange and green color for the cartoon yeasts throughout Figure 4C.

This is a very good remark. Our color code was not consistent with the genotype of the strains. We corrected this. Thank you for having detected this.

Minor Comment 1:

We now include in the Supplementary Information (Figure 9) a plot comparing in detail the data of Figure 2C and 2D regarding mNeonGreen expression, highlighting that optogenetic induction leads to expression levels slightly higher (around 10%) than pTDH3 after 4.5 hours of induction (timepoint #7).

The addition made by the authors has addressed my concern.

Thank you for having suggested this interesting addition.

Minor Comment 2:

We clarified that the OD plot for the multi-color culture is similar to the single-color cultures. We kept the sentence at the end of the legend “All bioreactor experiments presented in this figure were performed in parallel, the same day, with the custom bioreactor platform version” as it encompasses both the constitutive expression turbidostat cultures (Figure 2C) and light-induced gene expression characterization (Figure 2D).

The authors explanation has addressed my concern.

Minor Comment 3:

We agree and we have run these control experiments, confirming the stability of mCerulean expression in the WT strain, and also providing an estimate of mis-attribution error of our algorithm in real conditions: around 1/1000 cell was attributed a WT phenotype in mutant-only cultures and around 1/1000 cell was attributed a mutant phenotype in WT-only cultures. This means that the range of ratios between 0.01 to 100 is well suited for the reliable estimation of growth rate differences. All the corresponding data was in this suitable range with a comfortable margin. The corresponding data is now illustrated by two plots in Supplementary Information Figure 11, in the new section describing additional material regarding the three case studies.

For the OD-based consortium control experiment of Figure 4C, as discussed earlier, we have run extensive additional control and characterization experiments (Supplementary Information Figures 12 and 13).

The addition described by the authors has addressed this comment.

Good that we have clarified this issue.

Minor Comment 4:

We clarified that blue light was applied at time 0 and remained constantly on to induce and maintain the slow growth phenotype via expression of burdensome protein secretion.

The authors explanation has addressed this concern.

Minor Comment 5:

Cost comparisons for such platforms with different designs and capabilities (number of parallel cultures, temporal resolution, sample treatment), which are made by assembling several components of various origins (custom-made, commercial, mixed), is very difficult and somewhat arbitrary given that equipment prices are not public. We can still say that as of now, open-source, open-hardware components such as a Chi.Bio 8-bioreactors array (with 4 pumps per reactor and a controller board) costs US\$ 5895 from LabMaker. The cost of an Opentrons OT-2 pipetting robot with one multi-channel pipette is somewhat similar (US\$ 6800). We have added this information in the Supplementary Information within the adequate sections.

The authors have adequately addressed the concern.

Minor Comment 6:

We now include such a plot in Supplementary Information Figure 7B, also shown below.

The addition described by the authors has addressed this comment.

Thank you for having suggested this improvement.

Minor Comment 7:

We have improved the clarity of the plot (Supplementary Information Figure 8) and expanded the discussion of the expected impact of carry-over, which is very low. Briefly, less than one out of 600-800 cells will be left in the acquisition plate after the wash, and these cells won't be able to grow in-between timepoints because of lack of medium.

The addition described by the authors has addressed this comment.

Minor Comment 8:

We thank the reviewer for noting this. We found out the issue is at the level of the word to pdf conversion. We corrected the issue.

The author's explanation has addressed this concern.

Minor Comment 9:

We have added a detailed discussion on the topic cell size normalization in the section "Metrology and automated analysis of cytometry data from yeast cultures" of the Supplementary Information. Briefly, we acknowledge that while several works use the forward scatter for cell size normalization, other works use the side scatter, and we also recognize that probably no perfect, instrument- and condition-independent proxy for cell size can be computed from cytometry-based optical measurements. We thus used a data-driven approach based on the reduction of the cell-to-cell variability for constitutive gene expression to justify the use of the forward scatter. We also compared the shape of the FSC and SSC distributions with published distributions of yeast cell size obtained with Coulter counters and found that the FSC distribution resembled it more.

I appreciate the authors providing literature for both sides of the argument and clearly justifying their choice.

Thank you.

Minor Comment 10-11:

We have now edited a video highlighting key elements of a ReacSight platform and the key steps to run an experiment on a ReacSight platform. It also provides an additional view of the different components for automating cytometry acquisition with the OT-2. The video is available in the ReacSight gitlab repository: https://gitlab.inria.fr/InBio/Public/reacsight/blob/master/reacsight_presentation_video.mp4

I appreciate the additional effort to highlight the system with a video. I was unable to follow the link above but was able to find the video through the repository.

Thank you for letting us know about the link issue. We checked and the link is indeed not working from the outside. To avoid any accessibility issue, we will provide the video as supplementary material.

Minor Comment 12:

We have added files for cut instructions in open format (.svg and/or .dxf).

The addition described by the authors has addressed this comment.

Minor Comment 13:

We thank the reviewer for pointing this out. In fact, during testing we found that the `opentrons.robot.commands()` method of the Opentrons API v1 does not function as expected, preventing us to retrieve a list of elementary robot operations. However, because the instrument control does log all the commands sent to the OT2 Flask app as well as the responses of the Flask app, the platform will still generate a very detailed log because the robot actions exposed by the Flask app are very elementary. We have therefore removed the corresponding method in our Opentrons Flask app code.

The author's explanation has addressed this concern.

Minor Comment 14:

We thank the reviewer for pointing this out. We have changed the color scheme in Figure 3C. For Figure 3A, we chose not to repeat the Time label as it is indicated for the light intensity plot, but we added the ticks and tick labels to help the reader. We also did the same for Figure 3C, left.

I would like to thank the authors for addressing my concerns, however I have a few suggestions for consideration:

- In Figure 3C right, changing the color palette from rainbow made it less distracting. However, the color gradient is being used to visually encode time, but the x axis being timepoints does that independently without color. I suggest that each panel on the right be the same color as the corresponding median values represented in the left panel. For example, change the target = 0.2 panel to dark blue for all timepoints, and the target =0.8 panel to lightest blue for all timepoints.

Thank you for the suggestion. We did what you proposed.

- The authors could also consider replacing the right panel of Figure 3C with the new data presented in the supplement in response to Major Comment 1 (currently Supplementary Figure 10), and move the "gene expression in single cells" panel to the supplement. This would allow them to highlight their excellent data showing that "real time control of gene expression using light" was robust and reproducible. An argument could be made either way on the utility of the two pieces of data (single cell vs. the new data), so I would leave it to the authors to decide what they think is best!

We also like the new control experiment on robustness. We initially revised the figure to add this data in addition to the other one, but the figure became overly large when we added the novel panel on long term protein expression and secretion (panel 3d). In the end, we have chosen to keep the panel with protein distributions since otherwise this extremely important aspect does not appear much in the main text.

Typographical/attribution errors:

We now use the better term "waste funnel" consistently.

I appreciate the broad revision, however "thrash" still appears in Supplemental Figure 5.

Thank you for having spotted this. We hope we now have thrashed this poor terminology. 😊

2-7: We thank the reviewer for pointing these issues that we fixed. Indeed, sampling was manual in reference 6, we thank the reviewer for pointing this out and have adapted the manuscript accordingly.

9-15: We thank the reviewer for pointing these issues that we fixed. We are grateful to the reviewer for this very useful verification of our reference list. As mentioned above, we have updated how we cite reference 6.

The authors address our specific concerns, but we would encourage them to do another detailed verification for typos throughout the manuscript and supplement as they carefully review and revise the manuscripts for consistency from Major Comment 2. The following are non-exhaustive examples:

- Line 115: capitalize “Python”
- Line 185-186: “is” changed to “are” 2x
- Line 380, 381, 383: Formedium is inconsistently capitalized

We are sorry that a number of typos were still present in our first revision of the manuscript. We have once more proofread the main text and the supplement.

Reviewer #2 (Remarks to the Author):

The authors have addressed some of my concerns but from my perspective the study as presently described is more incremental than a substantial improvement on what has already been published.

Even if we do not view our contribution as incremental with respect to what has already been published, we acknowledge that different viewpoints exist, and we are glad that we addressed in our first revision at least some of the concerns raised by the reviewer in their previous review.

To make the case for a substantial improvement the authors need to either show an integration of multiple devices or an exemplar application.

In this second revision, we document the creation of another platform, combining a plate reader and a pipetting robot, and provide two additional applications, namely the investigation of gene expression stability over extended durations (120 hours, that is > 60 cell generations) when the protein is secreted and/or when the corresponding gene is present in multiple copies, and the effect of sustained or repeated antibiotic treatments on bacterial populations growing in different conditions. Altogether this corresponds to very significant additions to our previous work. We hope that we will convince the reviewer that our work is now worth publication. More details are provided below.

1. For example to show that ReacSight is a powerful generalisable framework to feedback from ex-situ measurements the authors need to demonstrate:

- (1) Multiple bioreactor systems can work with it (that’s already shown in the ms)
- (2) Multiple external measurement devices can be managed- e.g. at least a plate reader (or alternative instrument) application
- (3) Liquid handling robot can do more useful things than simple 20x dilution (which could be done without a robot) -> they argue enzymatic assays might be useful but so not show any data to support this.

2. Alternatively the authors could show a novel application that is enabled thanks to ex-situ measurement:

- (1) e.g. 5 day experiments as shown by Aditya et al., Nature Communications 2021 - as a difficult thing to do but can be achieved by the authors developments
- (2) the argument that pulsed (short) in-situ fluorescence measurements would indeed cause major accidental optogenetic actuation and therefore require ex-situ analysis would have to be supported by some data
- (3) the multiplexing/barcoding of mixed populations they suggest might be possible with 10s of channels should be attempted with more ambition (rather than just 2 populations)
- (4) perhaps offer a novel application

Regarding proposition 1.1 (ReacSight can be used to control multiple bioreactor systems), this was already demonstrated in the previous version of our manuscript, as rightfully noted by the reviewer,

since we provided results both with our custom bioreactor system and with the chi.bio bioreactor system.

Regarding proposition 1.2 (Multiple external measurement devices can be managed- e.g. at least a plate reader application), we have extended our work to demonstrate that ReacSight can be used to extend a plate reader with pipetting capabilities so as to implement complex reactive protocols. We thank the reviewer for this suggestion. More precisely, we provide two case studies with this platform. In the first one, we grow bacterial cell populations in microtiter plates and, when their optical densities reach a chosen threshold, we renew the media to keep cells growing in well-controlled conditions over extended durations (> 15 cell generations). We test the capacities of the cells to grow in different media (glucose with or without casamino acids) and with different antibiotic concentrations (sustained treatments). We observed that fast growing cells appeared to be less impacted by the treatments than slow growing ones, a somewhat counter-intuitive result. In the second case study, we grow cells in presence of various antibiotic concentrations, and apply a second treatment when the cell population reaches a chosen optical density (repeated treatments). Cell density is important in β -lactam treatments because of collective antibiotic tolerance. To document these new results, we have added a new section in the main text (ReacSight is a generic strategy: enhancing plate readers with pipetting capabilities, approx. 2 pages) that notably contains an additional figure (Figure 5).

We also followed proposition 2.1 (performing 5 days experiments). We constructed novel yeast strains that express a fluorescent protein, secreted or not, and present in multiple copies or not. We followed the expression levels over 120 hours, measuring the distribution of protein concentrations in the cell population every 2 hours. Cells also carry a reporter system for secretion stress. We have observed the emergence of complex behaviors (multimodality and non-monotonicity). These results are documented in a new panel of Figure 3, in two additional paragraphs (approx. $\frac{1}{2}$ page) and in two supplementary figures (Supplementary Figures 11 and 12). More globally, we are currently analyzing a small collection of yeast strains expressing various hard-to-secrete proteins for their effective secretion capabilities as a function of the production demand. These results will be part of another article.

Given that we proposed two novel applications, we naturally follow the last proposition too (2.4: offer a novel application).

Regarding the propositions 1.3 (performing more complex tasks with the pipetting robot) and 2.2 (developing more ambitious multiplexing/barcoding applications for mixed populations), we have chosen not to address them in this contribution. These two directions are both highly interesting (and in fact we have started to investigate them) but we feel that they are less relevant than other propositions made by the reviewer with respect to the specific topic of this contribution: demonstrating the capacity to build robust and reactive experimental platforms by connecting various pieces of equipment in a same programming environment.

Lastly, the point 2.2 (whether in-situ fluorescence measurements could cause unwanted optogenetic actuation) is interesting but is a comment with respect to a claim we made in our previous response to the reviewer. We were then providing four reasons for which *ex situ* sample analysis, as permitted by the use of a cytometer for example, can be beneficial in comparison to *in situ* measurements. Our most important point was that cytometry provides single-cell level data. This not only provides information on population heterogeneity but also opens exciting multiplexing capabilities, where many different genotypes and their associated phenotypes can be resolved from co-cultures thanks to fluorescence barcoding strategies. Therefore, resolving the issue raised by the reviewer is not directly relevant for this contribution. However, we will nevertheless investigate this question (we have the

chi.bio reactors and optogenetic strains) and document this point in the supplementary material if we have results before the publication of the paper (in Nature Communications, we hope).

Reviewers' Comments:

Reviewer #2:

Remarks to the Author:

Enhancing bioreactor arrays for automated measurements and reactive control with ReacSight

The authors followed several of my suggestions to strengthen the case for the novelty of their work, utility of their approach and as they wrote in their response "demonstrating the capacity to build robust and reactive experimental platforms by connecting various pieces of equipment in a same programming environment."

They addressed the major concerns sufficiently by adding new data to their manuscript:

1. An application leveraging a second measurement device – a plate reader in a continuous and reactive culture experiment in plate format. Presented in an additional Figure 5.

In my view this now sufficiently showcases the flexibility of their experimental and programming approach in managing multiple bioreactor / growth platforms in the context of multiple analytic devices.

2. An experiment that runs over 5 days – demonstrating the stability of the platform over time and emphasizing the need for automation in reactive continuous long-term experimentation. Presented in a new panel in Figure 3.

This is an excellent additional experiment, and the authors can consider two suggestions for the panel d in Figure 3/Suppl. Figure 11:

a. In case the experiment was performed with Chi.Bio bioreactors they might have time course data on total mNeonGreen levels (combining intracellular and secreted fluorescent reporter signal). It would be interesting to show how the platform can collect in parallel bulk and single-cell measurements for additional insights into productivity mechanisms. This could also be an opportunity for integrating flow cytometer and plate reader on the same platform (in future work).

b. The distribution graphs on the right of Figure 3/panel d are difficult to read.

Maybe sharing the time legend for the 4 graphs and providing stacked individual graphs for the distributions would make it easier to appreciate the trends the authors describe.

For the discussion sections the authors might consider referencing the 2022 publication from DeBenedictis et al., which applied their reactive experimental platform built with a python framework for continuous cultures on a Hamilton liquid handler with plate reader feedback functions to evolve novel biomolecules as a further powerful example of the approach the authors are advocating for.

DeBenedictis, E.A., Chory, E.J., Gretton, D.W. et al. Systematic molecular evolution enables robust biomolecule discovery. *Nat Methods* 19, 55–64 (2022). <https://doi.org/10.1038/s41592-021-01348-4>

Previous work already cited by the authors:

16.Chory, E. J., Gretton, D. W., DeBenedictis, E. A. & Esvelt, K. M. Enabling high-throughput biology with flexible open-source automation. *Mol. Syst. Biol.* 17, e9942 (2021).

So overall, in my view the revised manuscript is now suitable for publication in *Nature Communications*.

Reviewer #2 (Remarks to the Author):

The authors followed several of my suggestions to strengthen the case for the novelty of their work, utility of their approach and as they wrote in their response “demonstrating the capacity to build robust and reactive experimental platforms by connecting various pieces of equipment in a same programming environment.”

They addressed the major concerns sufficiently by adding new data to their manuscript:

1. An application leveraging a second measurement device – a plate reader in a continuous and reactive culture experiment in plate format. Presented in an additional Figure 5.

In my view this now sufficiently showcases the flexibility of their experimental and programming approach in managing multiple bioreactor / growth platforms in the context of multiple analytic devices.

2. An experiment that runs over 5 days – demonstrating the stability of the platform over time and emphasizing the need for automation in reactive continuous long-term experimentation. Presented in a new panel in Figure 3.

This is an excellent additional experiment, and the authors can consider two suggestions for the panel d in Figure 3/Suppl. Figure 11:

- a. In case the experiment was performed with Chi.Bio bioreactors they might have time course data on total mNeonGreen levels (combining intracellular and secreted fluorescent reporter signal). It would be interesting to show how the platform can collect in parallel bulk and single-cell measurements for additional insights into productivity mechanisms. This could also be an opportunity for integrating flow cytometer and plate reader on the same platform (in future work).

The experiment has been performed with our custom bioreactors that do not perform in situ fluorescence measurements. Therefore, we do not have this data.

- b. The distribution graphs on the right of Figure 3/panel d are difficult to read.

Maybe sharing the time legend for the 4 graphs and providing stacked individual graphs for the distributions would make it easier to appreciate the trends the authors describe.

We shared the time legend for the 4 graphs as suggested. However, we did not succeed to have a satisfying spatial arrangement of a stacked version of the 4 graphs.

For the discussion sections the authors might consider referencing the 2022 publication from DeBenedictis et al., which applied their reactive experimental platform built with a python framework for continuous cultures on a Hamilton liquid handler with plate reader feedback functions to evolve novel biomolecules as a further powerful example of the approach the authors are advocating for.

DeBenedictis, E.A., Chory, E.J., Gretton, D.W. et al. Systematic molecular evolution enables robust biomolecule discovery. *Nat Methods* 19, 55–64 (2022). <https://doi.org/10.1038/s41592-021-01348-4>

Previous work already cited by the authors:

16. Chory, E. J., Gretton, D. W., DeBenedictis, E. A. & Esvelt, K. M. Enabling high-throughput biology with flexible open-source automation. *Mol. Syst. Biol.* 17, e9942 (2021).

The work in the proposed citation is impressive. However, in the context of our contribution, we see the initial citation (Ref 16, also a very nice paper and by the same group) more appropriate.

So overall, in my view the revised manuscript is now suitable for publication in Nature Communications.

We thank the reviewer for their positive assessment of our work.